# Light-insensitive organic solar-powered amplifiers

Qiang Wu[1,8], Shijie Wang[1,8], Wei Gao[2,8], Zeng Chen[3], Jun Tao[4], Xinyue Song[5], Sen Yan[5], Haiming Zhu[3], Ke Zhou[1], Long Jiang[6] ✉, Xiaomin Xu[4] ✉, Alex K.-Y. Jen[2] ✉ & Wei Ma[1,7] ✉

Wearable bio-sensors using organic electrochemical transistors (OECTs) powered by flexible organic solar cells (OSCs) show promise for electrophysiological monitoring. However, single OECT bio-sensors face unstable outputs due to the limitations of OSCs under low-light conditions and poor energy autonomy. Here, we show a low-power self-powered physiological sensor employing a dual-OECTs configuration, connected in series and powered by the optimized OSCs, which shows more stable signal output and faster response compared to single OECT bio-sensors. Our devices employ the efficient and more stable OSCs to power amplifier by suppressing charge recombination and improving flexibility, thereby facilitating long-term, on-demand use. This integrated device can be attached to human-skin to stably monitor signals, including electrocardiograms, electromyograms and electrooculograms across a wide range of illumination intensities (500 lux-50,000 lux). The design offers a simple architecture for wearable low-power self-powered bio-sensors without external energy supplies/storage, highlighting their potential in real-time disease diagnosis and prevention scenarios.

As the demand for personalized health monitoring, sports tracking, and medical assistance rises, traditional rigid bioelectronic devices fall short in meeting the evolving needs for effective human integration[1,2]. The essential challenges confronting the next-generation wearable electronics include achieving high precision in data recording, ensuring energy autonomy, maintaining operational stability, and possessing high conformability[3–8]. The development of self-powered biosensors is critical in this realm, offering real-time and continuous monitoring of electrophysiological signals such as electrocardiography (ECG), electromyography (EMG), and electrooculography (EOG), paramount in contexts ranging from sports training to disease diagnosis. Research on organic electrochemical

transistors (OECTs) offers great promise in electrophysiological signals recording, where their key features include high signal amplification, low operating voltage, better biocompatibility, and compatibility with existing solution processes like printing techniques[9–11]. An OECT interfaced with human skin leverages electrophysiological signals as its gate voltage ($V_{GS}$), and an additional power supply as the source-drain voltage ($V_{DS}$), thereby transducing bioelectric signals into electrical outputs by modulating the source-drain current ($I_{DS}$). The stability of this signal output is notably prone to fluctuations in the power supply. Compared with the field effect transistors, OECTs exhibit much higher transconductance ($g_m$), which also inevitably increases $I_{DS}$ and power consumption[12]. Once the power

[1]State Key Laboratory for Mechanical Behavior of Materials, Xi'an Jiaotong University, Xi'an, China. [2]Department of Materials Science and Engineering, City University of Hong Kong, Kowloon, Hong Kong. [3]State Key Laboratory of Modern Optical Instrumentation, Key Laboratory of Excited State Materials of Zhejiang Province, Department of Chemistry, Zhejiang University, Hangzhou, China. [4]Institute of Materials Research, Shenzhen International Graduate School, Tsinghua University, Shenzhen, China. [5]School of Information and Communications Engineering, Xi'an Jiaotong University, Xi'an, China. [6]State Key Laboratory of Oil and Gas Equipment, CNPC Tubular Goods Research Institute, Xi'an, China. [7]XJTU Institute of Intelligent Optometry Equipment, Xi'an Jiaotong University, 710049 Xi'an, China. [8]These authors contributed equally: Qiang Wu, Shijie Wang, Wei Gao. ✉e-mail: jianglong003@cnpc.com.cn; xu.xiaomin@sz.tsinghua.edu.cn; alexjen@cityu.edu.hk; msewma@xjtu.edu.cn

supply is lower than demand, the amplification capability will be obviously restricted. Therefore, the circuits design with the lower-power demand that consistently provide stable signal output remains a barrier to the widespread adoption of wearable OECTs for electrophysiological monitoring.

Next-generation wearable biosensor devices utilizing OECTs must ensure not only stable signal output, but also energy autonomy for prolonged use. Recent advancements aim to address these requirements by pursuing self-powered OECT operations[4,5,13,14]. Ambient light, whether from sunlight or artificial indoor sources, provides a readily available energy source as the source-drain voltage and current input. For instance, Prof. Someya et al. reported an ultra-flexible, self-powered biosensor that integrates a single OECT with OSCs as the power source[4]. This device can measure biometric signals while being powered by OSCs under 20,000 lux of light-emitting-diode (LED) illumination. Note that wearable biosensors need to operate across a range of lighting conditions, from dimly lit indoor environments to bright outdoor settings, yet the performance of OSCs is highly dependent on lighting conditions (such as intensity and time). Under typical outdoor illuminating conditions, a high density of photo-generated charge carriers effectively mitigates electronic defects and trap states within the active and interfacial layers of OSCs, but their light stability is poor[15,16]. However, under indoor ambient lighting conditions (200–5000 lux), the same devices show a better light stability, but the density of photogenerated carriers is significantly reduced. This reduction impairs their capability to saturate defect sites, leading to increased charge recombination and noticeable power losses[17,18]. Consequently, documented OCSs generally exhibit reduced power, thus impacting energy autonomy and stable signal output for practical applications of the self-powered biosensors[19]. Specifically, in applications involving single OECT, the reduction of the power supply will confine the $I_{DS}$ and $g_m$ values of the device, thereby affecting the stability of electrophysiological signal outputs. Under unstable autonomous energy of OSCs, these fluctuations become more pronounced, potentially leading to device failure. Therefore, achieving the high power and better light stability of OSCs remain ongoing challenges.

In a systematic level, in order to address the above-mentioned challenges, supercapacitors or other energy storage devices have been integrated within the electronic circuits of wearable biosensors[6,7]. While complex power managers systems are adept at smoothing power output of solar cells under fluctuating conditions and maintaining the operational stability of wearable biosensors, their integration needs to be strategically dealt with to avoid increasing the size and complexity drastically. Innovations in system design must continuously aim at reducing the physical footprint and optimizing the energy flow management to bolster functionality across varying environmental settings. Therefore, morphological optimization is central to addressing the fundamental challenges of OSCs' power and operational stability, which are critically dependent on the control of the subtle nanometer-scale morphology. Among these approaches of the optimized morphology[20–23], the ternary concept features a third component added into the host active-layer system, which can suppress charge recombination to improve photovoltaic performance and otherwise be detrimental. Meanwhile, mechanical stability is also one of the important criteria for evaluating the application of OSCs in wearable self-powered biosensors, but it is often ignored. In addition, advancements in a system-level design of low-power OECT-based sensors are equally essential to ensure minimal power fluctuation, particularly in designs that eschew bulky energy storage modules. The above innovations are crucial for ensuring the long-term and reliable function of self-powered biosensors in diverse environmental conditions.

In this work, we report a wearable self-powered low-power unipolar amplifier designed to address the challenges of maintaining consistent performance in wearable biosensors, particularly under varying lighting conditions, completely eliminating the need for external energy storage modules. This low-power amplifier features a dual-OECTs configuration, connected in series and powered by the optimized OSCs, enabling the continuous and reliable monitoring of human electrophysiological signals under ambient lighting. To improve the power conversion efficiency (PCE) and stability of OSCs, we incorporated BS3TSe-4F as a guest acceptor and an insulating poly(aryl ether) (PAEN) into the PM6/BTP-eC9 host blend (LbL-Binary). The combination, termed LbL-Ternary/PAEN (PM6/PAEN/BTP-eC9:BS3TSe-4F), not only demonstrates superior photovoltaic performance with a PCE reaching 19.17%–surpassing the 18.02% PCE of LbL-Binary devices–but also enhances light and mechanical operational stabilities. For the biosensor component, the unipolar amplifier based on p(g2T-T) operates with a maximum gain of 93 (V/V) near $V_{in} = 0\,V$ and it shows low-power compared with the single OECT device. Of note is that the self-powered unipolar amplifier device based on the amplifier achieves a more stable signal output and rapid response, compared to a biosensor utilizing a single OECT unit. Furthermore, we demonstrated that this wearable self-powered sensor can be attached to the human skin, providing stable monitoring of physiological signals such as ECG, EMG, and EOG, across a wide range of illumination intensities, highlighting its potential application in real-time disease diagnosis and preventive healthcare scenarios.

## Results
### Design of the bio-electronic devices for wearable use
In order to meet the demands of power and stability (light and mechanical stabilities) of wearable biosensors for long-term usage, we selected PM6 as a donor, the small molecule non-fullerene acceptors (SM-NFAs)-BTP-eC9 and BS3TSe-4F as acceptors, and insulating PAEN material as an addition to construct OSCs with high PCE and better operation stability. The molecular structures and energy levels of PM6, BTP-eC9, and BS3TSe-4F are shown in Supplementary Figs. 1–2. The favorable energy levels of the above photovoltaic materials ensure efficient exciton dissociation and decrease charge recombination of OSCs[20,24–26]. Supplementary Fig. 3 displays the normalized absorption spectra of the investigated active layer pure materials and blending films. Thus, introducing BS3TSe-4F as a guest into the PM6:BTP-eC9 system provides the complementary and broadened absorption under the ambient light compared with that of the PM6:BTP-eC9 system. Another key factor in achieving better performance is the miscibility of the active layer materials, we carried out the surface energy measurements (Supplementary Fig. 4 and Supplementary Table 1). These related results imply that BTP-eC9 and BS3TSe-4F prefer to form an alloy-like phase in the ternary blend, making the pure phase of materials more crystalline and better morphology without damaging the scale of phase separation, consequently effectively decreasing charge recombination[24,27,28]. In addition, according to the previous reports[22,29], insulating PAEN material has highly twisted-stiff backbones without any side chains, so it can be introduced into the polymer: SM-NFAs system to enhance the mechanical stability of the active layer. Detailed information will be discussed below.

As shown on the left in Fig. 1a, the integrated biosensors of OSCs-powered single OECT can be attached to the human skin (gate) for collecting the current change of the drain to obtain electrophysiological signals[4]. The OSCs simultaneously act as both a current source and a voltage source, supplying adequate $I_{DS}$ and bias constant $V_{DS}$ during the OECT operation. In the applications involving single OECTs that work in linear region, small variations in the drain voltage can cause significant changes in the output drain current[30], thereby affecting the stability of electrophysiological signal outputs. Under unstable autonomous energy of OSCs, these fluctuations become more pronounced, potentially leading to device failure. Therefore, the short circuit current at maximum power point ($I_{SC, mpp}$)

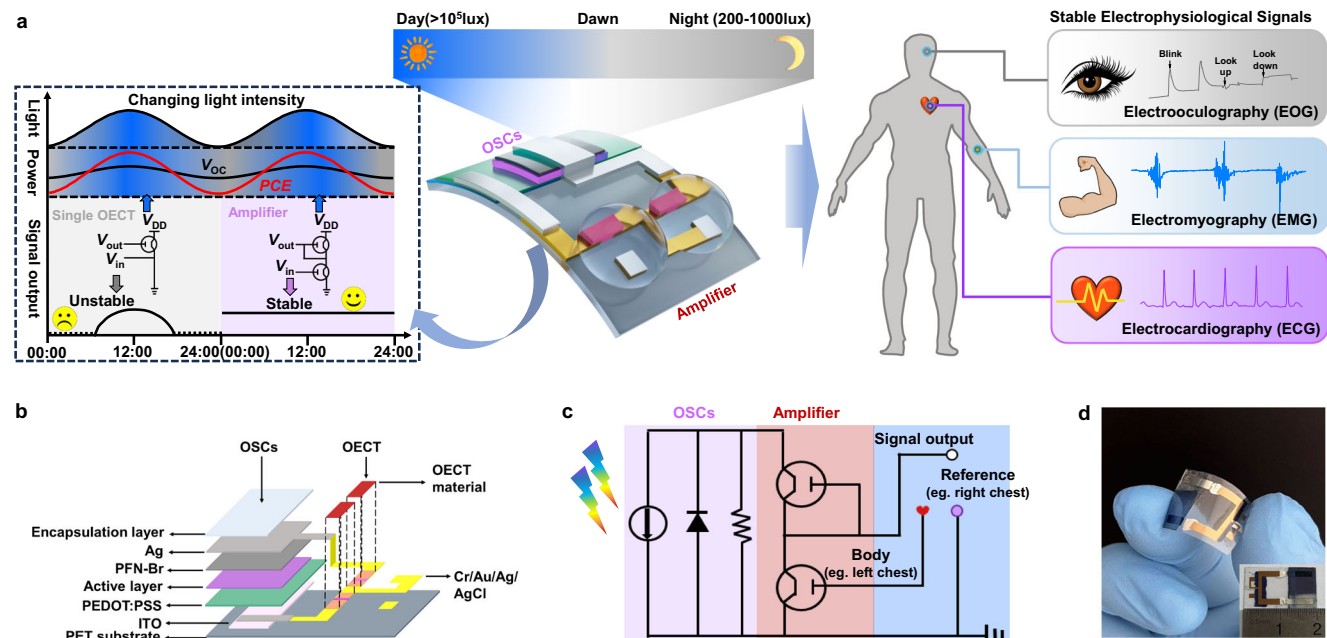

**Fig. 1 | Design of the wearable self-powered integrated bio-electronic device for monitoring electrophysiological signal of the human body. a** Schematic illustration of the wearable self-powered unipolar amplifier consists of two independent OECT in series using OSCs for monitoring electrophysiological signal under different light intensity conditions. **b** Layered illustration of components in the self-powered integrated bio-electronic device. **c** Circuit diagram of the amplifier and OSCs. **d** Optical image of the self-powered integrated bio-electronic device.

of the OSCs under low light illumination must be at least equal to or more than the $I_{DS}$ required under forward bias to meet the enough power for stable operation, assuming the open-circuit voltage at maximum power point ($V_{OC, mpp}$) of OSCs as $V_{DS}$ is negligible changes. In this work, we applied a low-power unipolar amplifier, which features a dual-OECTs configuration, connected in series. For this kind of circuit, the trip point ($V_{TP}$) does not depend on the $V_{DD}$ that supplied by the OSC. When the input is low (e.g., input voltage ($V_{in}$) = 0) the source-gate voltage ($V_{SG}$) applied to the driver OECT is large ($V_{SG} = V_{DD} - V_{in}$), and the output voltage ($V_{out}$) is close to the supply voltages ($V_{DD}$). By increasing $V_{in}$, the pull-up can be weaker and a sharp transition of $V_{out}$ from $V_{DD}$ to 0 V is displayed when $V_{in}$ reaches the $V_{TP}$. Further increasing $V_{in}$ will reduce the $V_{SG}$, and $V_{out}$ will be close to 0 V. To fix the $V_{TP}$ near 0 V, p(g2T-T) was selected as the channel materials of the OECTs due to its appropriate threshold voltage ($V_{th}$). As a result, this low-power amplifier powered by the OSCs based on the PM6/PAEN/BTP-eC9:BS3TSe-4F system enables the continuous and reliable monitoring of human electrophysiological signals (such as ECG, EMG, and EOG) under ambient lighting, and then transmits the information to the electronic product. Figure 1b presents an expanded exploded schematic illustration of a wearable self-powered biosensor device. Among this integrated device, an OECT-based unipolar amplifier was used as a low-power biosensor for the recording of electrophysiological signals.

The circuit diagram of the self-powered biosensor shows the harvested energy flow and provides a concise overview of the device operation (Fig. 1c). The cathode of the OSCs is connected to the supply voltage terminal ($V_{DD}$) of the amplifier, and the anode of the OSCs is connected to the other end of the amplifier as a reference electrode. The $V_{in}$ terminal of the amplifier is connected to the human body's organs or tissues to be measured (e.g., left chest), and the reference electrode is connected to the human body's reference part (e.g., right chest). When light is irradiated, the $V_{out}$ of the amplifier outputs a voltage signal and transmits it to the mobile phone, thereby recording human electrophysiological signals. A view of the wearable self-powered bio-integrated device of is presented in Fig. 1d, which shows

that the overall size of the self-powered bio-integrated device is approximately 1.5 cm × 2.0 cm and can be comfortably adhered to the skin.

## OSCs design and characterization

To enhance the PCE and stability of OSCs, we prepared these corresponding OSCs devices by using a conventional architecture of indium tin oxide (ITO)/poly(3,4ethylene dioxythiophene):poly(styrenesulfonate) (PEDOT:PSS)/Active layer/poly(9,9-bis[6(N,N,N trimethylammonium) hexyl]fluorene-alt-co-1,4-phenylene-bromide) (PFN-Br)/Ag to evaluate the photovoltaic performance and operation stability (light and mechanical stabilities). The detailed device fabrication procedures are provided in the Supplementary Information. The current density versus voltage ($J$–$V$) curves of the optimized systems are taken under AM 1.5 G illumination, as shown in Supplementary Fig. 5, and the corresponding photovoltaic parameters are summarized in Supplementary Table 2. With BS3TSe-4F added into the PM6:BTP-eC9 binary blends based on the bulk heterojunction (BHJ) counterpart, the PM6:BTP-eC9:BS3TSe-4F ternary device with an optimal weight ratio of 1:1:0.2, the optimized device shows a PCE as high as 18.35% in comparison with those of the binary OSCs. According to the pervious results[31–35], it is found that the sequential layer-by-layer (LbL) deposition technique printing film can effectively reduce charge recombination and improve mechanical stability compared to its BHJ counterpart. Taking into account the consideration of the advantages of the LbL deposition technique, we performed the PM6/BTP-eC9 binary device (LbL-Binary), which can afford higher efficiency (up to 18.02%) than the BHJ counterpart. Impressively, the optimized ratio of BS3TSe-4F and BTP-eC9 device (LbL-Ternary) exhibited the highest PCE of 19.17% by the LbL method (Fig. 2a and Table 1).

As a power supply module for wearable biosensors, it is also necessary to consider the mechanical properties of OSCs[36–38]. According to the previous reported[22,29,39], the incorporation of PAEN can improve the mechanical robustness of the active layer films due to the strong chain entanglement effect. Unlike the previous processing method, we deposited PAEN on top of PM6 to form this structure of

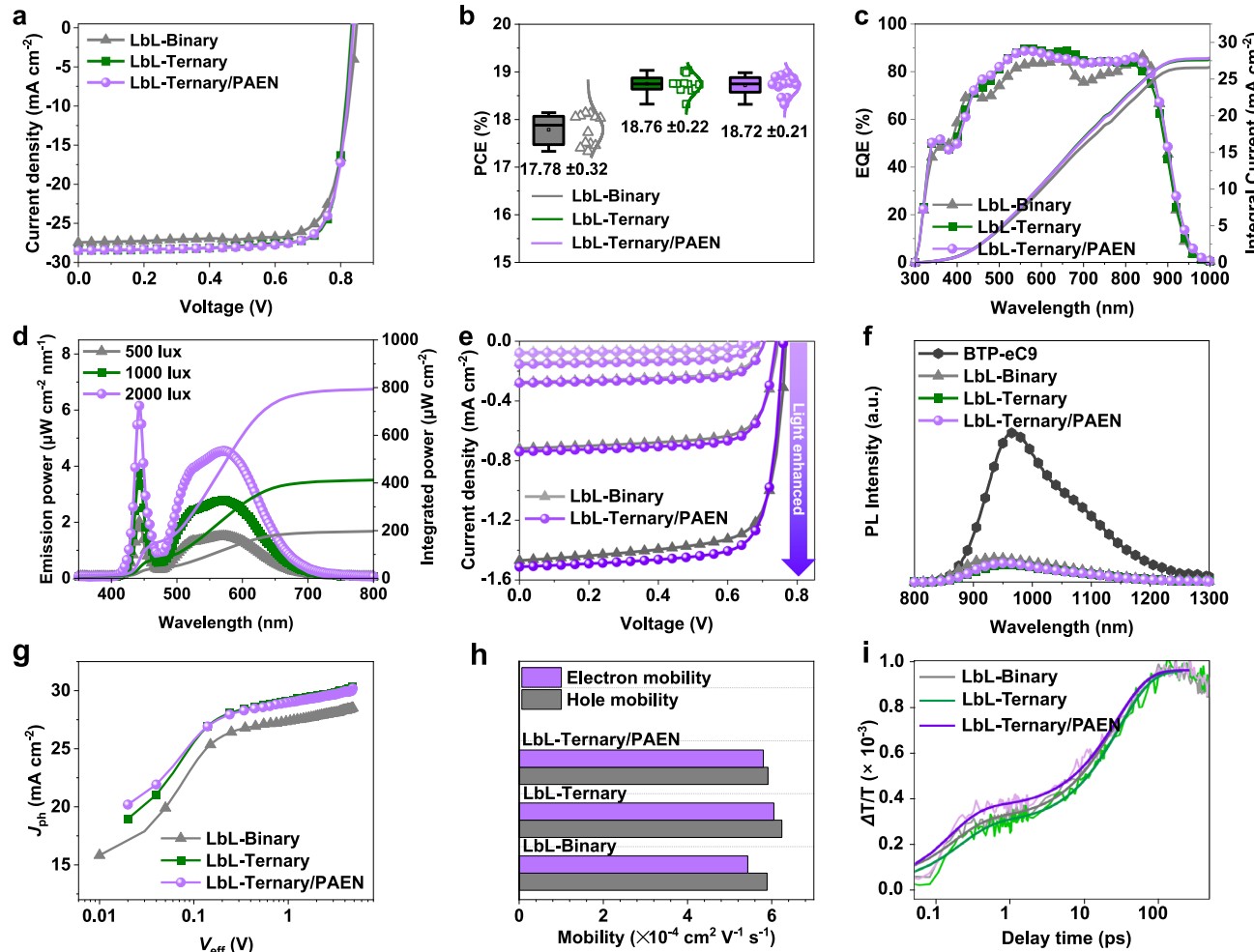

**Fig. 2 | Performance and photoelectric characteristics of the OSCs devices.**
**a** The $J-V$ curves of the three OSCs devices measured under 1 sun illumination.
**b** Statistic PCE of OSCs with different active layer system (12 individual devices and expressed as mean ± s.d.). **c** EQE spectra of the three OSCs devices. **d** Emission power and integrated power spectra of the LED at different light intensities (500, 1000, and 2000 lux). **e** The $J-V$ curves of the LbL-Binary and LbL-Ternary/PAEN devices under different indoor light intensities. **f** PL spectra of the pristine acceptor (BTP-eC9) and the corresponding films. **g** Characteristics of the photocurrent density versus effective voltage ($J_{ph}-V_{eff}$). **h** The hole and electron mobilities of the corresponding OSCs devices. **i** Normalized TA kinetics and their fitting curves of the corresponding OSCs devices.

**Table 1 | Photovoltaic parameters of the binary and the optimized ternary OPV cells based on the LbL deposition technique**

| Sample | | $V_{OC}$ (V) | $J_{SC}$ (mA cm$^{-2}$) | $J_{SC, EQE}$ (mA cm$^{-2}$) | FF (%) | PCE[a] (%) |
|---|---|---|---|---|---|---|
| PM6/BTP-eC9 | LbL-Binary | 0.849 | 27.45 | 26.64 | 77.31 | 18.02 (17.78 ± 0.32) |
| PM6/BTP-eC9:BS3TSe-4F | LbL-Ternary | 0.840 | 28.44 | 27.69 | 80.29 | 19.17 (18.76 ± 0.22) |
| PM6/PAEN/BTP-eC9:BS3TSe-4F | LbL-Ternary/PAEN | 0.838 | 28.48 | 27.89 | 79.53 | 18.98 (18.72 ± 0.21) |

[a]The average PCE values with standard deviations were obtained from 12 individual devices.

PM6/PAEN/BTP-eC9:BS3TSe-4F(LbL-Ternary/PAEN). Impressively, the PCE of the LbL-Ternary/PAEN does not differ from that of the device without PAEN (LbL-Ternary). The statistical photovoltaic metrics obtained from the three systems are depicted in Fig. 2b, indicating the good reproducibility of the corresponding device performance. As plotted in Fig. 2c, relevant EQE measurements of the optimized devices were carried out, which agree well with those obtained from the $J-V$ measurements within 5% mismatches. In addition, we further focused on the indoor photovoltaic performance of the LbL-Binary and LbL-Ternary/PAEN devices. As shown in Fig. 2d, e and Supplementary Table 3, the output power ($P_{out}$) of the LbL-Binary and the LbL-Ternary/PAEN devices achieved 63.01 mW cm$^{-2}$ and 67.86 mW cm$^{-2}$ under a 1000 lux LED light, respectively. Thus, the high $P_{out}$ of LbL-Ternary/

PAEN device inspired us to initially try to provide enough power for biosensors based on OECT device.

Next, to investigate the charge recombination probabilities of the host binary and the optimized ternary devices, we have studied the exciton dynamics, the charge generation, transport, and the collection processes in the corresponding optimized devices by employing steady-state and transient spectroscopic techniques. As presented in Fig. 2f, g and Supplementary Table 4, the quenching efficiencies and $P_{E,T}$ values of the LbL-Ternary/PAEN device are higher than other systems. Meanwhile, the more balanced hole- and electron-only mobilities ($\mu_h/\mu_e = 1.03$ or 1.02) indicates the more efficient charge transport properties in the ternary system, compared with the binary device ($\mu_h/\mu_e = 1.08$) (Fig. 2f, g and Supplementary Table 4). As provided in

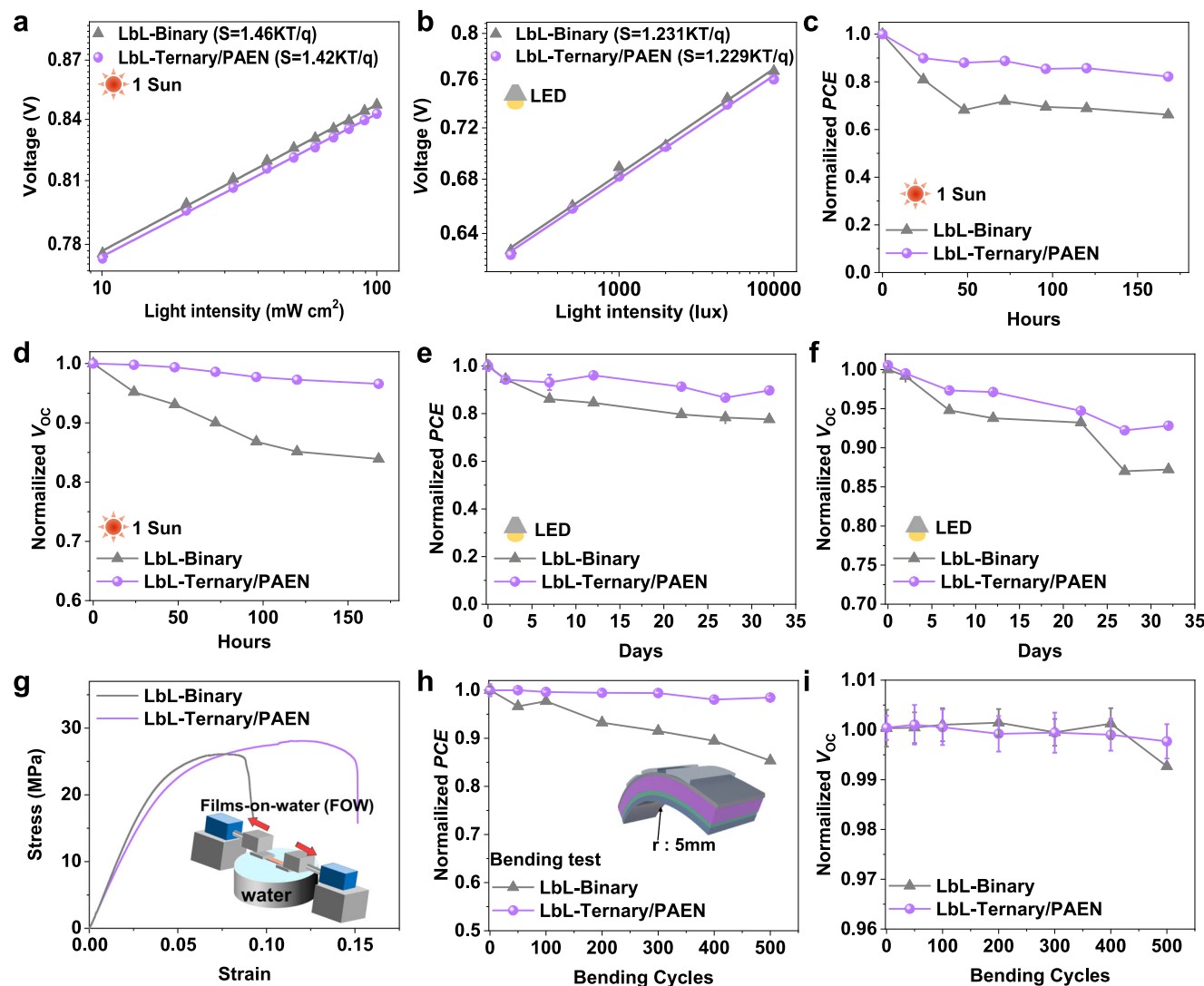

**Fig. 3 | Stability characteristics of the PCE and $V_{OC}$ of the corresponding OSCs devices. a** $V_{OC}$ of the corresponding OSCs devices as a function of illuminance under one sun illumination. **b** $V_{OC}$ of the corresponding OSCs devices as a function of illuminance under LED illumination. Variation of normalized PCE (**c**) and $V_{OC}$ (**d**) losses over 168 h under one sun illumination for the corresponding OSCs devices based on the unencapsulated devices measured in a $N_2$-filled glovebox. Variation of normalized PCE (**e**) and $V_{OC}$ (**f**) losses over 32 days under LED illumination for the corresponding OSCs devices (3 individual devices and expressed as mean ± s.d.) based on the unencapsulated devices measured in a $N_2$-filled glovebox. **g** The stress-strain curves of the corresponding films, and the insert shows the schematic image of the tensile tester setup for floated active-layer films. Normalized PCE (**h**) and $V_{OC}$ (**i**) for the relevant flexible devices (3 individual devices and expressed as mean ± s.d.) with different bending cycles, and the inset shows the schematic image of bending cycles with a bending radius of 5 mm.

Supplementary Fig. 6, the correlations between $J_{SC}$ and light intensity ($P_{light}$) were studied. The similar α values close to unity indicate negligible bimolecular recombination during sweep-out in the corresponding devices[40–43]. In addition, we have also investigated the exciton relaxation dynamics of the corresponding devices using femtosecond transient absorption spectroscopy (TAS)[25,31,44]. As depicted in Fig. 2i and Supplementary Table 5, the hole-transfer process (HT) in the corresponding optimized films shows a fast component with $τ_1$ of -0.21/-0.18/-0.15 ps, and a slow component $τ_2$ of 27.8/25.8/27.2 ps, respectively. In addition, the exciton diffusion mediated hole transfer process ($A_2$) account for 67, 66, and 60% on LbL-Binary, LbL-Ternary and LbL-Ternary/PAEN are more than the interfaces transfer process ($A_1$) account, indicating the significant role of exciton diffusion in the LbL blend microstructure. Remarkably, the LbL-Ternary/PAEN film exhibits charge generation and decay dynamics similar to those of LbL-Ternary film, which indicates the existence of insulating PAEN in active layers does not inhibit charge generation or induce additional charge recombination, in good agreement with the pervious results[29]. Based on

the abovementioned device performance and physical mechanisms (Fig. 2f–h and Supplementary Figs. 6–9) of the host binary and optimized ternary devices, the introduction of the BS3TSe-4F acceptor and PAEN into the LbL-Binary system can accelerate charge transfer and inhibit charge recombination, and consequently achieve a higher PCE.

Undoubtedly, a thorough morphological study may be helpful to understand the above-mentioned differences of charge recombination probability of the corresponding active layer films. We firstly focus on the position of PAEN as the third component in the LbL-Ternary/PAEN film. Of note is that the C/O ratios of PM6, BTP-eC9, BS3TSe-4F, and PAEN are 25.5, 32.33, 32.33, and 12.0, respectively, so we can estimate C/O ratio from depth analysis X-ray photoelectron spectroscopy to investigate the vertical separation of PAEN based on the LbL-Ternary/PAEN film. The C/O ratio of the LbL-Ternary film is uniformly distributed throughout the whole active layer. On the contrary, the peak intensity of the O atom is significantly enhanced, primarily due to the introduction of PAEN into the LbL-Ternary film (Supplementary Fig. 10, normalized by the intensity of the C atom). These results further

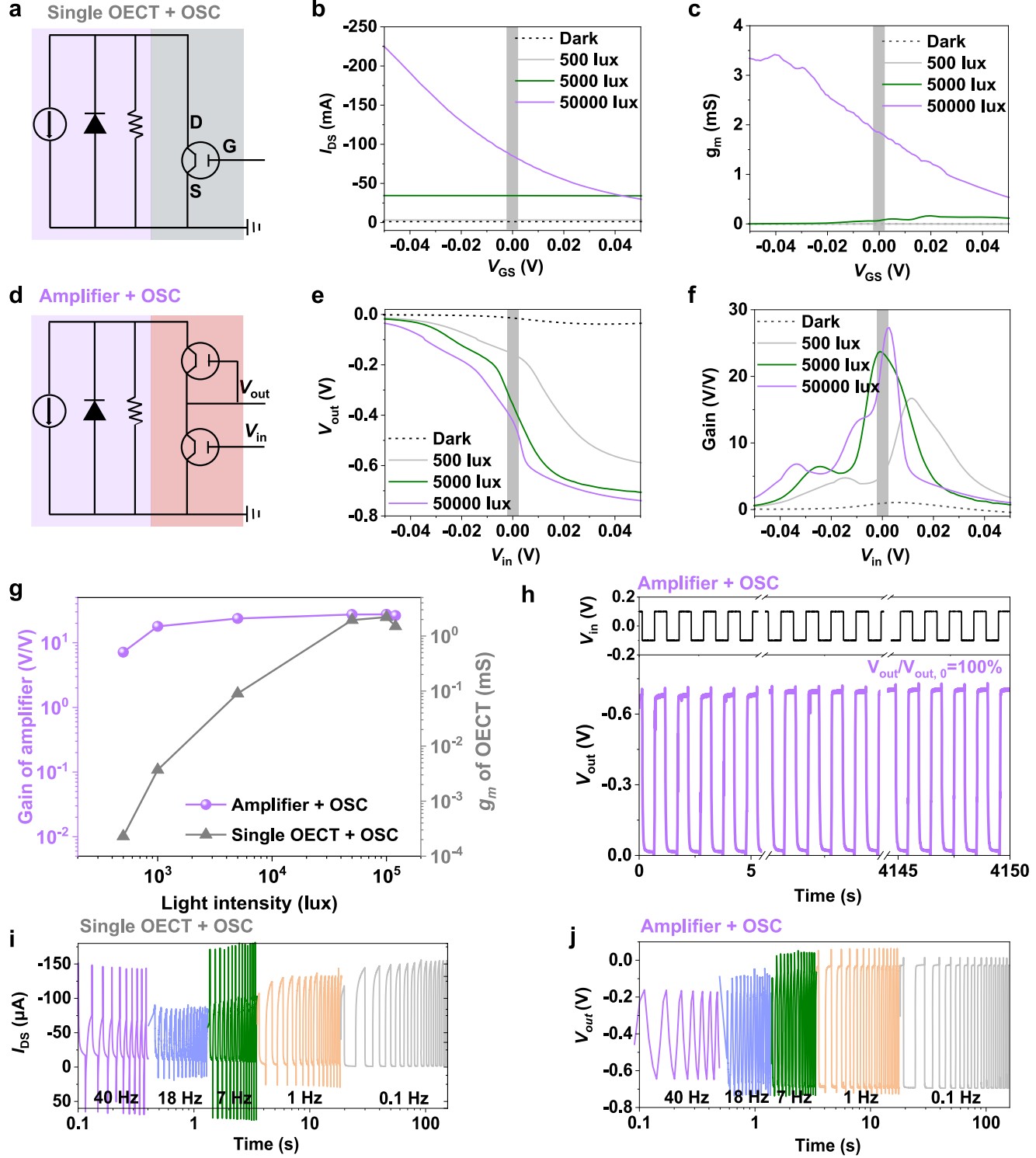

**Fig. 4 | Performance of OSCs powered single-OECT and amplifier.** Circuit diagram of the OSCs powered OECT (**a**) and amplifier (**d**). Transfer curves (**b**) and related transconductance (**c**) of the OSCs powered OECT at different light intensity. Voltage transfer curves (**e**) and related gain (**f**) of the OSCs powered amplifier at different light intensity. **g** The transconductance of the OSCs powered OECT and gain of OSCs powered amplifier as functions of light intensity. **h** Operation stability of the OSCs powered amplifier with an input frequency of 1 Hz. The light intensity used here is 50,000 lux. Transient response of the OSCs powered OECT (**i**) and amplifier (**j**) with different input frequency.

confirmed the insertion of PAEN into LbL-Ternary film. From atomic force microscopy (AFM) as shown in Supplementary Fig. 11, the LbL-Binary, LbL-Ternary and LbL-Ternary/PAEN films exhibit similar mean-square surface roughness (RMS), which exhibit a very smooth surface with the macroscopic phase separation. Since AFM has limitations in resolving the blend microstructures, 2D grazing incident wide-angle X-ray scattering (GIWAXS) were further employed to define morphological properties (Supplementary Fig. 12), and the relevant crystallographic parameters of these films are summarized in Supplementary Table 6. As shown in Supplementary Fig. 12, these optimized films show preferential face-on orientation relative to the substrate due to the obvious (010) signal in the out-of-plane (OOP) direction, and the

almost similar lamellar peaks at $q_z$ 0.303 Å$^{-1}$ for the in-plane of donor (IP (100)) (d-spacing 20.74 Å) and 0.391 Å$^{-1}$ for the in-plane of acceptor (IP (100)) (d-spacing 16.07 Å). In addition, the calculated crystallite coherence lengths (CCL) values are 31.57 Å for LbL-Binary, 33.60 Å for LbL-Ternary, and 33.60 Å for LbL-Ternary/PAEN in the OOP direction, respectively. These results imply the introduction of BS3TSe-4F and PAEN are not change the d-spacing, but effectively increased crystallinity and reduced defect state of the morphology, thus achieve the better performance.

## Light and mechanical stabilities of OSCs for wearable use

Apart from the better performance via eliminating the charge recombination probability and accelerating the charge transfer, the operation stabilities are important for the application of a wearable self-powered device. Therefore, we firstly carried out the LbL-Binary device and LbL-Ternary/PAEN device for achieving a quantitative comparison. Figure 3a, b shows the light-intensity-dependent $V_{OC}$ changes of the LbL-Binary and LbL-Ternary/PAEN devices under varying light intensity based on the solar simulator and LED. Note of that the relationship between $V_{OC}$ and $P_{light}$ can be used to describe the trap-assisted recombination through nkT/q[45–47]. The LbL-Binary device shows a large slope of 1.46 kT/q under the solar simulator condition and 1.231 kT/q under LED illuminating, while the LbL-Ternary/PAEN device exhibit a small slope of 1.42 kT/q under the solar simulator condition and 1.229 kT/q under LED illuminating. These results indicated the LbL-Ternary/PAEN device have less trap-assisted recombination than the LbL-Binary device, which is conducive to achieving high power output under different lighting conditions. And we then investigated the PCE and $V_{OC}$ losses by light stability and bending of devices measurements based on the different usage conditions. Normalized PCE and $V_{OC}$ as a function of illumination time under one sun conditions are provided in Fig. 3c, d. The LbL-Ternary/PAEN device maintained 82.22% and 96.6% of its initial PCE and $V_{OC}$ after continuous illumination for 168 h (one week), whereas the LbL-Binary device retained 66.22% and 83.9%, respectively. Impressively, the LbL-Ternary/PAEN device maintained 89.73% and 92.8% of its initial PCE and $V_{OC}$ after continuous LED 1000 lux illumination for 32 days at room temperature (Fig. 3e, f). Thanks to the small change in PCE and $V_{OC}$ under varying light intensity conditions, the optimized LbL-Ternary/PAEN device can provide a stable power input to drive biosensors. In addition, considering that the failure characteristic of the ultimate stress for human skin is approximately 3 MPa ± 1.5 MPa, and the overall longitudinal strain of human skin was less than 9.5% ± 1.9%. To develop a quantitative understanding of the mechanical stabilities of the optimized films, we further employed a pseudo-free-standing tensile test on a water surface[21,48,49]. As shown in Fig. 3g and Supplementary Table 7, the crack-onset strain (COS) of the optimized LbL-Ternary/PAEN ternary film (12.3%) is higher than that (7.0%) of the LbL-Binary film. The better mechanical stability of the higher COS film was also confirmed by the calculation of toughness. In order to further verify the correctness of our strategy, we prepared the flexible devices of LbL-Binary and LbL-Ternary/PAEN system. As shown in Supplementary Fig. 13a and Supplementary Table 8, the LbL-Ternary/PAEN device based on PET substrate achieved 18% for PCE and 0.823 V for $V_{OC}$, while the LbL-Binary devices achieved 17.04% for PCE and 0.839 V for $V_{OC}$. Furthermore, their performances depending on bending cycles were investigated. Figure 3h, i exhibits the normalized PCE and $V_{OC}$ of the LbL-Binary and LbL-Ternary/PAEN flexible devices as a function of bending cycles with a radius of 5 mm. After 500 cycles of bending tests, the LbL-Ternary/PAEN device exhibits a more stable performance compared to the LbL-Binary device. These results indicate that PAEN can significantly increase the mechanical stability of the relevant active layer, which agree with the pervious results[22,39]. Notably, Supplementary Fig. 13b shows the plots of PCE values versus years for the reported efficient all-PSCs in literatures and the corresponding statistical parameters are listed in Supplementary Table 9. The LbL-Ternary/PAEN system is the one of the highest PCE of flexible OSCs. In this work, the LbL-Ternary/PAEN system achieved a higher performance and operation stability, which are encouraged to provide enough power to drive the biosensors based on the OECT for all-weather monitoring.

## Performance of OSCs powered single-OECT and amplifier

Leveraging the better PCE and operation stability of the LbL-Ternary/PAEN OSCs, we fabricated a bio-electronic device integrating an OSC-powered OECT for monitoring human electrophysiological signal. We firstly evaluated the performance of the amplifier and single OECT based on p(g2T-T) powered by the Source Meter Unit (SMU). Supplementary Fig. 14 shows the transfer curve of p(g2T-T)-based OECT. The OECT exhibits an on/off ratio of 10$^4$ and a $V_{th}$ of 0.006 V, the transconductance ($g_m$) and subthreshold swing (SS) at 0 V is 1.4 mS and 76 mV/dec, respectively. The low SS, accompanied with ideal $V_{th}$, are promising to equip the amplifier with high gain at $V_{in}$ = 0 V. As shown in Supplementary Fig. 15, the resulting amplifier shows a $V_{TP}$ near 0 V, which facilitates accurate monitoring of the electrophysiological signals due to their potential near 0 V. The gain value ranges from 46 to 93 V/V with negligible $V_{TP}$ shift when the $V_{DD}$ changes from 0.5 V to 0.8 V, indicating the amplification capability will not be obviously limited by reducing the supply voltage, which agrees with the previous results[50,51]. Compared to single OECT, the power and current consumption is much lower for the unipolar amplifier with much higher total resistance, which can be better supplied by the optimized OSCs. Second, the $V_{OC}$ (i.e., $V_{DD}$) of the optimized OSCs, serving as another factor that limits the performance of amplifier, shows much lower dependence on light intensity than $J_{SC}$. Finally, the influence of $V_{OC}$ fluctuation on $V_{TP}$ and the gain value of the unipolar amplifier is also minimal.

To validate the superiority of our configuration design, we fabricated the amplifier and a single OECT powered by the optimized OSCs. A design drawing is presented for an integrated bio-electronic device comprising an OSC-drive amplifier or a single OECT based on a single substrate (Supplementary Fig. 16). As shown in Fig. 4b, c and Supplementary Fig. 17a, b, the $g_m$ near 0 V is greatly limited by the light intensity based on the single OECT. Specifically, when the light intensity is lower than 5000 lux, the $g_m$ is lower than 0.1 mS—over 20 times lower than the value (2.2 mS) acquired at high light intensity of 100,000 lux. This limitation is mainly because the photocurrent at the maximum power point (MPP) supplied by OSCs ($J_{MPP}$ ~ 25.4 μA, $V_{MPP}$ = 0.64 V, $P_{MPP}$ ~ 16.3 μW at 5000 lux) cannot satisfied the current-hungry single OECT ($I_{DS}$ ~ 45.4 μA, $P$ ~ 27.2 μW at $V_{GS}$ = 0 V, $V_{DS}$ = −0.6 V) in a dim surrounding environment. In sharp contrast, as shown in Fig. 4e, f and Supplementary Fig. 17c, d, the OSC powered amplifier can still keep a considerable gain value (~10 V/V) near 0 V even under low light intensity of 500 lux. Further increasing the light intensity leads to a high and stable gain that ranges from 18 to 26 V/V at $V_{in}$ = 0 V. This mainly originates from the low demand power and current of amplifier ($I_{DD}$ = 16.5 μA, $P$ ~ 9.9 μW at $V_{in}$ = 0 V, $V_{DD}$ = −0.6 V) (Supplementary Fig. 18). Thus, we have demonstrated the high adaptability of our self-powered bioelectronics across different surrounding environment.

As another key factor for practical applications, the operation stability of OSCs powered amplifier is also assessed. Figure 4h shows the $V_{out}$ response of the amplifier upon ±100 mV $V_{in}$ pulses at 50,000 lux. Surprisingly, the fast and stable response can still be maintained after 4150 s operation in air, with no performance deterioration ($V_{out}$/$V_{out,0}$ = 100%), indicating that both the power and sensing module exhibits extraordinary stability. We ascribe the stability to two aspects: (1) enhanced photostability achieved by LbL-Ternary/PAEN OSCs and (2) relatively high stability of the p(g2T-T)-based OECT during the cyclic redox process. The single OECT powered by the same OSCs maintained 88% of its initial peak on-state current after 4150 s light illumination (50,000 lux) and electrochemical cycling, probably due to the redox side reactions or material structure evolution

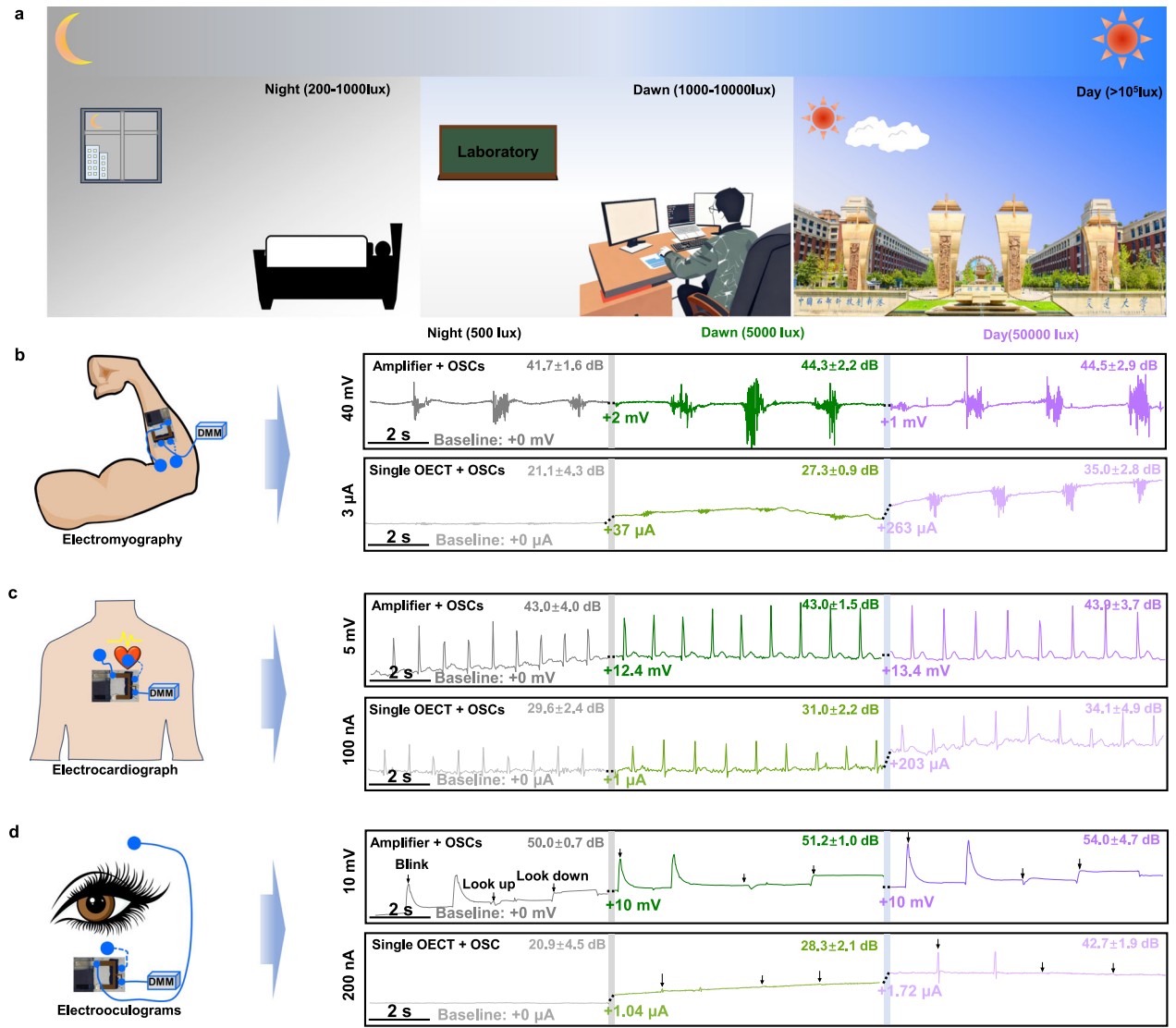

**Fig. 5 | On-body evaluation of the self-powered bio-electronic device for the recording of electrophysiological signals (EMG, ECG, and EOG) under different light intensity conditions. a** Schematic representation of the environment with different light intensities. Illustration of the self-powered bio-electronic device attached on the muscle for the monitoring EMG (**b**), ECG (**c**), and EOG (**d**) under different light intensity conditions (The dotted line represents the direct contact between the electrode and the human body (heart, muscles, eyes) to transmit signals.).

(Supplementary Fig. 19). This result indicates that the amplifier can be less sensitive to device performance degradation. It is well-known that the intrinsically sluggish ion transport will cause the slow response of OECTs, leading to the distortion of the electrophysiological signals, especially in the high-frequency domain. This problem can be alleviated by replacing the single OECT with amplifier. As shown in Fig. 4i, j, the transient response of the OSC powered OECT attenuates obviously along with the increase of the input frequency, only 40% of the $\Delta I_{DS}$ ($\Delta I_{DS} = I_{DS,max} - I_{DS,min}$) can be maintained when the frequency increases from 0.1 to 40 Hz. As comparison, 70% of the $\Delta V_{out}$ ($\Delta V_{out} = V_{out,max} - V_{out,min}$) can be maintained by using the OSC powered amplifier. In sum, the stable amplification capability, better operation stability, as well as the fast response make our OSCs-powered amplifier promising for around-the-clock electrophysiological signal monitoring.

## On-body device evaluation for monitoring electrophysiological signal

Human activities, such as muscle contraction, heartbeat, and eye movement, generate electrophysiological signals that are transmitted

through the internal environment and ultimately to the human skin. Before the on-body recoding, we first assessed the biocompatibility of the integrated device using the L-929 cell-culture experiment ("Method" section). As shown in Supplementary Fig. 20, both OECT channel and PDMS encapsuled OSC part show higher cell density and mitochondrial activity than control samples (bare glass). This indicates that the good biocompatibility and safe cell-device interfaces can be achieved using our all-organic integrated system. Taking advantage of the robust performance and high biocompatibility of this self-powered biosensors, we then carried out electrophysiological signal recording under different surrounding environment. As illustrated in Fig. 5a, it is potential for monitoring the electrophysiological signals from different parts of the human body (e.g., heart, muscles, and eyes) across a wide range of illumination intensities by applying integrated amplifier devices. Specifically, the $V_{in}$ terminal of the amplifier is connected to the human body's organs or tissues (e.g., left chest), and the reference electrode is connected to the reference body's part (e.g., right chest). Upon light exposure, the $V_{out}$ terminal of the amplifier outputs a voltage signal, which is transmitted via Bluetooth to a mobile phone for electrophysiological signal recording (Supplementary Movies 1, 2 and Supplementary Fig. 21).

As shown in Fig. 5c, taking the EMG recording as an example, the OSCs-powered amplifier consistently records EMG signal under LED lighting conditions ranging from 500 to 50,000 lux (e.g., day or night illumination). In contrast, OSC-powered single OECTs exhibit severe baseline fluctuations of the drain current with changes in light intensity (≤5000 lux) and can capture signals under daylight (50,000 lux). This phenomenon can be attributed to the fact that the power of OSCs can better satisfy the power demand of the amplifier than a single OECT, especially when the surrounding light intensity is low (Supplementary Fig. 18b). Similarly, both ECG and EOG signals are stably recorded under varying light intensity. For the EOG signal, it is worth mentioning that the OSCs-powered amplifier can accurately detect these electrophysiological signals of the eye blinking, looking up or down, manifesting its potential use in human-machine interface (HMI) or Virtual Reality (VR) technology in the future.

To quantitatively evaluate the signal quality of integrated devices under varying illumination conditions, we conducted signal-to-noise ratio (SNR) analyses ("Methods" section). As illustrated in Fig. 5c–e and Supplementary Table 10, the OSC-powered amplifier demonstrates superior performance in electrophysiological recordings, exhibiting both higher SNR values and reduced light-intensity dependency compared to single OECT devices. For instance, in EMG recordings, the amplifier maintains a high SNR of $44.5 \pm 2.9$ dB at 50,000 lux (day conditions), and even at 500 lux (night conditions), it achieves a commendable SNR of $41.7 \pm 1.6$ dB. In contrast, the SNR of signals recorded by a single OECT significantly decreases from $35.0 \pm 2.8$ dB to $21.1 \pm 4.3$ dB under the same conditions. The optimized OSC-powered amplifier consistently delivered high SNR and signal quality across multiple electrophysiological measurement scenarios—whether under day, dawn, or night conditions—substantially outperforming the single OECT configurations. To validate practical implementation, we extensively evaluated its long-term electrical and mechanical stability, along with other non-ideal factors such as motion artifacts. Supplementary Fig. 22 illustrates the device's long-term on-body stability over one week (168 h), revealing only a slight decrease in gain from 42 to 32 V/V on the first day, after which it stabilized. Additionally, the device was employed for long-term ECG monitoring, successfully capturing high-SNR ECG signals from day 1 to day 7 (Supplementary Fig. 22d). Mechanical stability, a critical factor for flexible devices, was also tested. As depicted in Supplementary Fig. 22, the device exhibited no significant performance degradation after 500 bending cycles (radius: 5 mm). These findings underscore the better long-term stability of our integrated device, attributable to the robust OECT amplifier and OSC (Fig. 3c–i) modules. Additional noise characterization studies addressed potential degradation mechanisms and motion artefacts, with the system showing particular resistance to such interference factors. The SNR of ECG signals recorded by the OSC-powered amplifier was assessed during both rest and jogging states (Supplementary Fig. 24a). Regarding device degradation, Supplementary Fig. 24b, c shows no obvious SNR decay after one week (168 h), regardless of activity state, confirming the device's stability and minimal noise generation during monitoring. For motion artifacts, while the SNR decreased from $39.9 \pm 0.5$ dB to $33.0 \pm 10.5$ dB during jogging, the ECG signal remained clear, with an SNR ($32.7 \pm 1.8$ dB) still surpassing that of a single OECT at rest ($30.7 \pm 2.2$ dB). This demonstrates the device's versatility and applicability in diverse scenarios, including prolonged physical activities. In summary, our research establishes that the optimized OSC-powered amplifier delivers stable and reliable electrophysiological signal recording with minimal light intensity dependence, paving the way for self-powered wearable biosensors suitable for indoor, point-of-care health monitoring, and Internet of Things applications.

## Discussion

We reported a wearable low-power self-powered biosensor that features a unipolar amplifier consisting of a dual-OECTs connected in series and powered by the optimized OSC. This integrated system can effectively monitor human electrophysiological signals, including ECG, EMG, and EOG, providing more stable signal output and reliable operation. To optimize the power supply of OSCs, we introduced BS3TSe-4F and PAEN into the PM6/BTP-eC9 system to suppress charge recombination and enhance molecular entanglement, thus achieving the higher PCE and the better operation stability for on-body wear. As for the sensor component, the unipolar amplifier based on p(g2T-T) operates with a stable gain value near $V_{in} = 0$ V, and it shows low-power than the biosensor based on the single OECT. As a result, the amplifier device powered by the optimized OSCs maintained a more stable signal output and faster response than the biosensors based on the single OECT, across a wide range of illumination intensity, demonstrating the feasibility of our system design. However, the response speed of this self-powered biosensors still need further improvement to enlarge the amplitude of elelctrophysiological signals. Future work for this technology will focus on miniaturizing the integrated device, and then part of the features functionality of this technology could be integrated with HMI or VR technology to enable gesture control for human or robotic prosthetics.

## Methods

### Materials
PM6, BTP-eC9, and PFN-Br were purchased from Solarmer Materials Inc. BS3TSe-4F was provided by Prof. Alex K.-Y. Jen lab[52]. p(g2T-T) was synthesized in our laboratory according the pervious paper[53]. Insulating PAEN material was provided by Dr. Jianhua Han. The aqueous dispersion of PEDOT:PSS was purchased from Xi'an Yuri Solar Co., Ltd. Chloroform was dried and distilled from appropriate drying agents prior to use.

### Characterization of photovoltaic materials
**Cyclic voltammetry (CV) measurements.** Electrochemical properties were studied by cyclic voltammetry (CV), which was performed on a CS350H electrochemcial workstation with a three-electrode system in a 0.1 M $Bu_4NPF_6$ acetonitrile solution at a scan rate of 20 mV s$^{-1}$. Glassy carbon disc coated with the sample film was used as the working electrode. A Pt wire was used as the counter electrode and Ag/AgCl was used as the reference electrode.

**Ultraviolet-visible-near-IR spectroscopy measurements.** UV–Vis absorption was measured using a PerkinElmer Lambda 1050 Spectrometer.

### Fabrication and measurement of OSCs devices
Solar cells were fabricated in the configuration of the traditional sandwich structure with an indium tin oxide (ITO, Advanced Election Technology Co. Ltd.) glass positive electrode and a perylene diimide functionalized with poly(9,9-bis[6(N,N,N trimethylammonium) hexyl] fluorene-alt-co-1,4-phenylene-bromide) (PFN-Br)/Ag negative electrode. The ITO-based substrates were pre-cleaned in an ultrasonic bath of detergent, deionized water, acetone and isopropanol, and UV-treated in ultraviolet-ozone chamber (Jelight Company, USA) for 15 min. A thin layer of PEDOT:PSS (poly(3,4-ethylene dioxythiophene): poly(styrene sulfonate)) was filtered through a 0.45 μm poly(tetrafluoroethylene) (PTFE) filter and spin-coated at 5500 rpm for 30 s on the ITO substrate. Subsequently, PEDOT:PSS film was baked at 150 °C for 15 min in the air, and the thickness of the PEDOT:PSS layer is approximately 30 nm. For the BHJ structure, the PM6:BTP-eC9:BS3TSe-4F blends (1:1.2:0, 1:1.1:0.1, 1:1.0:0.2, 1:0.9:0.3, and 1:0:1.2 weight ratio) were dissolved in CF solvents (16 mg mL$^{-1}$ in total) and stirred for at least 6 h. Then, these blend solutions were spin-coated on the PEDOT:PSS layer to form active layers. For the PM6/BTP-eC9:BS3TSe-4F (acceptor ratio: 1.2:0 and 1.0:0.2) device based on sequential LbL methods, the neat PM6 layer (10 mg mL$^{-1}$, CF) was spin-

coated onto the top of PEDOT:PSS, and then BTP-eC9:BS3TSe-4F (1.2:0 and 1.0:0.2) solution in CF (12 mg mL$^{-1}$; DIO:0.5 Vol%) was quickly spin-coated on top of PM6 layer. For the PM6/PAEN/BTP-eC9:BS3TSe-4F (1.2:0 and 1.0:0.2) device, the PAEN materials was dissolved in N-methylpyrrolidone (NMP) (0.5 mg mL$^{-1}$) and then spin-coated on top of PM6 layer. After acceptor layer deposition, these films were annealed at 100 °C during 10 min. The thickness of the photoactive layer is measured by a surface profiler. A PFN-Br layer via a solution concentration of 0.5 mg mL$^{-1}$ was deposited the top of active layer. Finally, a top Ag electrode of 100 nm thickness was evaporated in vacuum onto the cathode buffer layer at a pressure of $5 \times 10^{-6}$ mbar. The typical active area of the investigated devices was 4 mm$^2$.

For the fabrication of flexible OSCs, the PET/ITO substrates were cleaned with acetone and isopropanol (IPA) and ready for use. A ~30-nm-thick PEDOT:PSS layer was then spin-coated on the clean substrates followed by thermal annealing at 120 °C for 10 min in air to remove the residual water. The substrates were then transferred into a glovebox. And then, the flexible OSCs solar cells were prepared according to the method of glass substrate preparation.

The current-voltage characteristics of the solar cells were measured under AM 1.5 G irradiation on an Enli Solar simulator (100 mW cm$^{-2}$). Before each test, the solar simulator was calibrated with a standard single-crystal Si solar cell (made by Enli Technology Co., Ltd., Taiwan, calibrated by The National Institute of Metrology (NIM) of China). Short circuit currents under AM1.5 G (100 mW cm$^{-2}$) conditions were estimated from the spectral response and convolution with the solar spectrum. The external quantum efficiency was measured by a Solar Cell Spectral Response Measurement System QE-R3011 (Enli Technology Co., Ltd.). A calibrated silicon detector was used to determine the absolute photosensitivity at different wavelengths.

### Morphology measurements
Surface energy measurements were performed using a water or CH$_2$I$_2$ contact angle measurement system, and the surface energy was calculated using the equation of state. Using Wu's method, the corresponding surface energy (γ) was calculated based on the contact angle date of the different neat films.

AFM images were obtained by using Nano Wizard 4 atomic force microscopy (KRUSS-DSA100S, Germany) to observe the surface morphologies of the different films deposited on glass substrates.

### 2D GIWAXS measurements.
GIWAXS Characterization: GIWAXS measurements were performed at beamline 7.3.3 at the Advanced Light Source (America). Samples were prepared on Si substrates using identical blend solutions as those used in devices. The 10 keV X-ray beam was incident at a grazing angle of 0.11–0.15°, selected to maximize the scattering intensity from the samples. The scattered X-rays were detected using a Dectris Pilatus 2 M photon counting detector.

### Physical tests
**Photoluminescence (PL) measurements.** PL data were collected using a Edinburgh Instruments-FLS1000 Spectrometer (U.K.). The PL excitation wavelength was set to 639 nm.

**Transient absorption spectroscopy (TAS).** For femtosecond TAS, the fundamental output from Yb: KGW laser (1030 nm, 220 fs Gaussian fit, 100 kHz, Light Conversion Ltd) was separated to two light beam. One was introduced to NOPA (ORPHEUS-N, Light Conversion Ltd) to produce a certain wavelength for pump beam (550 nm/750 nm, below 7.5 μJ/cm$^2$/pulse, 30 fs pulse duration), the other was focused onto a YAG plate to generate white light continuum as probe beam. The pump and probe overlapped on the sample at a small angle less than 10°. The transmitted probe light from the sample was collected by a linear CCD array.

### Space charge limited current (SCLC) measurements.
Single carrier devices were fabricated, and the dark current-voltage characteristics measured and analyzed in the space charge limited current regime following the references. The structure of hole only devices was Glass/ITO/PEDOT:PSS/active layer/MoO$_3$ (15 nm)/Ag (100 nm). For the electron only devices, the structure was Glass/ITO/ZnO/active layer/PFN-Br/Ag (100 nm). Mobilities were extracted by fitting the current density-voltage curves using the Mott–Gurney relationship.

### Fabrication and characterization of OECT-based amplifier
For the fabrication of OECT-based amplifier, the glass substrates were pre-cleaned in an ultrasonic bath of detergent, deionized water, acetone and isopropanol, and UV-treated in a ultraviolet-ozone chamber (Jelight Company, USA) for 15 min. And then the 5 nm Cr and 50 nm Au were deposited onto the substrates by thermal evaporation with a shadow mask, which serves as the drain and source electrode. The p(g2T-T) was dissolved in CF (5 mg mL$^{-1}$) and then spin-coated onto the Glass/Cr/Au substrates with a speed of 1500 rpm. The resulting channel dimension of OECT is $W = 1000$ μm, $L = 50$ μm, $d = 70$ nm

The transfer/output curves and transient reponse behaviors of the OECTs were measured using a two-channel SMU (Keithley, 2602B SMU). For the single OECT, the first channel supplied $V_{DS}$ and measured $I_{DS}$ while the second channel provided $V_{GS}$. For amplifier, two linked 2602B SMUs were used to supply the $V_{DD}$, $V_{in}$, and monitor the $V_{out}$. Kickstart software was used to control the instruments. Ag/AgCl ink was droped onto the Au electrode to form the Ag/AgCl gates. 1 M NaCl aqueaous solution was used as the electrolyte.

### Device integration
The flexible OSCs solar cells and amplifier were prepared as described. And then, the cathode of the OSCs is connected to the supply voltage terminal ($V_{DD}$) of the amplifier, and the anode of the OSCs is connected to the other end of the amplifier as a reference electrode (Fig. 1c). The pre-patterned ITO coated glass or PET substrates, the mask of evaporate electrode based on the OSCs and the mask of evaporate electrode based on amplifier were disigned as shown in Supplementary Fig. 16.

### Physiological signal recording
The ECG, EMG, and EOG recordings were performed on a healthy volunteer. The participant in this experiment was male (33 years old) and provided informed consent. As the study was not designed to investigate gender-based differences, no gender-specific experimental design was employed. Before the recording, the skin at the recording sites was cleaned using soap and 50% v/v isopropyl alcohol. Next, commercial ECG ion gel was smeared onto the skin to act as the electrolyte. The exposed OECT channel was directly attached to the skin. The other OECT of the amplifier, which is also gated by ion gel, was insulated with skin using polyimide tape. Meanwhile, a medical gel-assisted Ag/AgCl electrode was pasted on the reference sites of the skin, and then connected to the reference electrode (cathode) of the integrated devices using a copper cable. Physiological signal was used as the $V_{GS}$ of the single OECT and $V_{in}$ of the amplifier. The signals were recorded by monitoring the $I_{DS}$ of the single OECT or $V_{out}$ of the amplifier using a digital multimeter (Tectronix DMM). The SNR of electrophysical was calculated using the following equation: SNR (dB) = 20log (Peak/σ).

$$SNB(dB) = 20 \log(Peak/\sigma)$$

Where "Peak" denotes the average peak voltage/current amplitude of the signals. "σ" denotes the standard deviation of the noise voltage/current at 0 V input. All procedures involving human research participants were conducted in accordance with the experimental protocol approved by the Ethics Committee of the First Affiliated Hospital of Xi'an Jiaotong University (the protocols of 2022–1203)

## Stability measurements

**Light stability test of OSCs.** We measured the light stability of the unencapsulated devices in this study, under different light source illumination (one sun and LED 1000 lux) at room temperature in $N_2$-filled glovebox for some days ($H_2O < 1$ ppm, $O_2 < 1$ ppm, 30 °C). The emission spectra and light intensity of the fluorescent lamps and the LED light sources were measured by a Fiber Optics Spectrometer (QE65 Pro, Ocean Optics).

**Light stability test of OSC powered amplifier.** We measured the light stability of the encapsulated OSC powered amplifier in air environment with high light intensity (50,000 lux LED) at room temperature. During the test, square waves with frequency of 1 Hz and amplitude of 0.1 V were used as the input.

**Bending stability test of the flexible devices.** For the evaluation of the device's flexibility, the OSCs and OECTs were repeatable bended on a cylinder with a radius of 5 mm and characterized its optoelectronic properties afterwards.

**Pseudo free-standing tensile test.** For the tensile testing specimen, the active layers were spin-coated onto the PEDOT:PSS/glass substrate. The active layer specimen with a size of $1.2 \times 0.5$ cm was prepared by using a cutting plotter. To float the specimen on the water surface, water was allowed to penetrate into the PEDOT:PSS layer. Subsequently, PEDOT:PSS was dissolved, and the active layer was delaminated from the glass substrate. By performing this process at the water surface, the floating active layer specimen could be obtained. Specimen gripping was achieved by attaching PDMS-coated Al grips on the specimen gripping areas using van der Waals adhesion. The tensile test was performed by a linear stage with a strain rate of 0.001 mm/s. All of tensile tests were carried out under the ambient conditions (Temperature ~25 °C, relative humidity (RH)~30 %).

## Cell experiment

**Cell culture.** NCTC clone 929 (L-929) cells were cultured incubated in a humidified atmosphere incubator with 5% $CO_2$ at 37 °C, and the culture medium was refreshed every 2 days throughout the incubation period. The bare glass, PDMS encapsuled OSC and p(g2T-T) channel on glass substrates were sterilized and placed centrally in 12-well plates, subsequently cells were seeded on each sample at a density of 104 cells/mL and incubated for 1, 3, and 5 days. In this experiment, the cells (104 cells/mL) cultured on bare glass samples were used as control. At each time point, the cells-adhered samples were rinsed twice with PBS and transferred into new 12-well plates. 500 μL 3-(4,5-Dimethylthiazol-2-yl)−2,5-diphenyltetrazolium bromide (MTT) solution (5 mg/mL MTT in PBS) was added to each well with continuous culture at 37 °C for 4 h. After removal of the solution, 500 μL DMSO was added into each well and oscillated for 10 min. Finally, 100 μL of the resultant solution was pipetted from each well and transferred to a new 96-well plate, and the absorbance was measured at 470 nm using the Multiscan GO microreader (Thermo Scientific, USA). Three different samples from each group were used at each time point. The statistical significance is analysed using One-way analyses of variance (ANOVA) and then a least-significant-difference (LSD) post hoc test is used to determine the level of significance.

**Live/dead staining.** Live/dead staining using the Live/Dead Viability/Cytotoxicity Kit (Molecular Probes, Invitrogen, France) was performed to identify viable and nonviable cells on the samples. At the end of each time period, the cell-adhered samples were washed three times using PBS followed by the addition of 500 μL of PBS containing ethidium-homodimer-1 (4 μM) and calcein-AM (2 μM) to each well prior to incubation at 37 °C for 25 min. The fluorescence-stained cells were analyzed using a fluorescence microscope (Nikon, Japan) for the collection of images. The obtained images were used to quantify the number of live/dead cells on the samples using ImageJ software. Three different samples from each group were used at each time point. The statistical significance is analyzed using One-way analyses of variance (ANOVA) and then a LSD post hoc test is used to determine the level of significance.

## The use of self-powered wearable integrated devices

The ECG measurement setup was built around a circuit board, which housed the key components: an ADS1292 chip for signal acquisition, singlechip (STM32l051) and a E104-bt5005A Bluetooth low-energy module for communication (3.8 V Li-ion electronic power supply). The connection between this board and the self-powered biosensor is established through flexible electrodes and connecting wires (Supplementary Fig. 21). ECG signals are streamed in real time to a mobile device (Redmi 14C) for display within a dedicated app. The participant in this experiment was male (33 years old) and provided informed consent. As the study was not designed to investigate gender-based differences, no gender-specific experimental design was employed. All procedures involving human research participants were conducted in accordance with the experimental protocol approved by the Ethics Committee of the First Affiliated Hospital of Xi'an Jiaotong University (the protocols of 2022–1203)

## Reporting summary

Further information on research design is available in the Nature Portfolio Reporting Summary linked to this article.

## Data availability

All data supporting the findings of this study are available within the main text and Supplementary Information file. Source data are provided with this paper.

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

## Acknowledgements

W. Ma acknowledges the support from the National Natural Science Foundation of China (W2411049), National Key Research and Development Program of China (2024YFA120824). K. Zhou thanks the support from the National Natural Science Foundation of China (52173023). X-ray data were acquired at beamlines 7.3.3 and 11.0.1.2 at the Advanced Light Source, which is supported by the Director, Office of Science, Office of Basic Energy Sciences, of the U.S. Department of Energy under Contract No. DE-AC02-05CH11231. All authors thank Dr. Eric Schaible and Dr. Chenhui Zhu at beamline 7.3.3 and Dr. Cheng Wang at beamline 11.0.1.2 for assistance with data acquisition. Q. Wu thanks the support from the National Postdoctoral Program for Innovative Talent (BX20220249), the

China Postdoctoral Science Foundation (2023M742762), Shaanxi Post-doctoral Science Foundation project (2023BSHYDZZ46). S. Wang thanks support from the National Natural Science Foundation of China (523B2033) and National Postdoctoral Program for Innovative Talents (BX20240282). L. Jiang thanks support from the Scientific Research and Technology Development Project of China National Petroleum Cor-poration (Grant No. 2024ZG50).

## Author contributions

Q.W., S.-J.W., and W.M. conceived and developed the ideas. Q.W. designed the experiments and performed OSCs devices fabrication. S.-J.W. performed single OECT and amplifier devices fabrication. W.G. and A.K.-Y.J. provided BS3TSe-4F acceptor. Z.C. and H.-M.Z. did the femto-second transient absorption spectroscopy. X.-Y.S. and S.Y. built the wireless communication. J.T. and X.-M.X. helped us make flexible devi-ces. Q.W., S.-J.W., J.L., K.Z., and W.M. wrote the manuscript. All authors participated in the discussions for manuscript preparation.

## Competing interests

The authors declare no competing interests.
