## [Transparent Peer Review File · Nature Communications]

Light-Insensitive Organic Solar-Powered Amplifiers

Corresponding Author: Professor Wei Ma

Version 0:

Reviewer comments:

Reviewer #1

(Remarks to the Author)

In this manuscript, Ma and co-workers reported a wearable self-powered bio-sensor that features a unipolar amplifier consisting of a dual-OECTs and powered by the OSC. Regarding the power supply unit, the author incorporated BS3TSe-4F as guest acceptor and an insulating poly(aryl ether) (PAEN) into the PM6/BTP-eC9, which system achieved better power and operational stabilities. For the bio-sensor module, the unipolar amplifier based on p(g2T-T) shows low-power and better performance compared with the single OECT device. As a result, the author demonstrated that this self-powered integrated device shows more stable physiological signal output and faster response compared to bio-sensors based on single OECT, across a wide range of illumination intensities (500 lux-50,000 lux).

In this integrated device level, the authors highlighted the simple architecture for wearable low-power self-powered bio-sensors without external energy supplies/storage, which are crucial for ensuring the long-term and reliable function of self-powered biosensors in diverse environmental conditions. I think this manuscript is scientifically interesting and technological important and would be one of the highly important research article in this journal. However, there are some critical concerns that must be addressed, and the authors need to carefully revise the manuscript.

1.The author selected insulating poly(aryl ether) (PAEN) material as addition to construct the better performance of OSCs. I suspect that the film absorption of PAEN has a certain effect on the device performance, such as JSC, but I have not found the relevant data. I suggest that the authors should also provide the film absorption of PAEN.

2.PAEN is a type of an insulating polymer material, so why is device performance not affected? Please explain it.

3.The authors should provide the standard deviation of PCE, such as Table S3.

4.As OECT sensor are also illuminated by light together with OSC, the authors should demonstrate the channel current of OECT sensor will be significantly influenced by light.

5.In Figure 4h, the authors showed that the amplifier can be very stable during the long-term operation. The stability of the single OECT and related discussion should also be provided here.

6.This manuscript is coherently written, but some mistakes need further correction, as follows: 1) In abstract section, this sentence "We show that this integrated self-power bio-sensor....., across a wide range of (500 lux-50,000 lux) illumination intensities." is corrected as "We show that this integrated self-power bio-sensor....., across a wide range of illumination intensities (500 lux-50,000 lux)." 2) Photostability or light stability of these words, which is described stability. Please keep the manuscript consistent. 3) "single OECT" is not "single OECTs". Please carefully check and correct all errors in this manuscript.

Reviewer #2

(Remarks to the Author)

The manuscript presents wearable organic electrochemical transistor (OECT)-based biosensors powered by organic semiconductors (OSCs). The study highlights a dual-OECT configuration that achieves stable signal output and faster response times for electrophysiological signal recording. Additionally, the authors report flexible OSCs with a power conversion efficiency (PCE) of 19.04% and robust operational stability. Despite these achievements, the work falls short in demonstrating a fully integrated wearable electronic system. The lack of comprehensive system-level demonstrations limits the practical applicability and impact of the proposed technology. Furthermore, the manuscript does not present significant advancements in materials, fabrication processes, device configurations, or performance metrics for either OSC or OECT technologies. These limitations prevent the work from meeting the innovation and impact criteria expected for publication in Nature Communications. Regrettably, the reviewer cannot recommend this manuscript for publication in Nature Communications. Detailed comments and constructive feedback are provided below for the authors' consideration.

1. The presented work lacks an innovative approach to device integration, a critical aspect for wearable technologies. The fabrication of OSCs on a plastic substrate and OECTs on a glass substrate using disparate techniques such as evaporation and spin-coating contradicts the concept of a seamlessly integrated wearable device. A single-substrate approach is essential for the high-level integration expected in modern wearable electronics, which the manuscript fails to deliver. The lack of an actual system-level demonstration, including the integration of OSCs, OECTs, and NFC components, makes it difficult to assess the feasibility of the entire system as a wearable device. Images or schematics showing the complete integrated system are necessary to evaluate the practical implementation of the described technologies.
2. The manuscript does not clarify the power management between the OSCs and the NFC chip. It is crucial to specify whether the NFC is self-powered by the OSCs or relies on external power sources, as this impacts the device's autonomy and practicality in wearable applications.
3. Concerns about the physical and operational scalability of the organic electrochemical transistors, given their relatively large size, are not addressed. The rationale behind the chosen device dimensions and the impact on overall device performance and application potential should be critically analyzed.
4. The experimental results, particularly those related to on-body applications, lack comprehensive quantitative analysis. Data on the signal-to-noise ratio and detailed frequency analysis are necessary to substantiate the claims of effective physiological signal monitoring. There is no sufficient evidence provided on the biocompatibility and long-term stability of the device when in contact with human skin. These are crucial factors for any wearable technology, especially those intended for continuous health monitoring.
5. Authors need to adequately address the stability of the device under real-world conditions, including its mechanical and electrical stability under repeated stress and strain, which are typical in wearable scenarios.
6. Noise considerations, particularly those arising from motion artifacts, are not discussed. The stability of device adhesion on a PET substrate during actual use and its effect on signal integrity need thorough examination and robust testing data.

Reviewer #3

(Remarks to the Author)

In this work, they have developed a low-power self-powered physiological sensor that features a dual-OECTs configuration, connected in series and powered by the optimized champion OSCs. The devices employ the efficient and more stable OSCs to power amplifier by suppressing charge recombination and improving flexibility, facilitating long-term, on-demand use. In addition, the integrated self-powered biosensor can be attached to human skin for stable monitoring of physiological signals, including ECG, EMG, and EEG, over a wide range of (500 lux-50,000 lux) illumination intensities. There are some important issues need to clarify/discuss before it can be considered for publication in Nature Communication. These are:

1. In this work, the advantages of PM6, (SM-NFAs)-BTP-eC9 and BS3TSe-4F being selected are not shown.
2. The authors believe that "introducing BS3TSe-4F as guest into PM6:BTPeC9 system provides the complementary and broadened absorption under the ambient light compared with that of the PM6:BTP-eC9 system." However, this opinion is inconsistent with the results in Figure S3. Apparently, BS3TSe-4F and BTPeC9 have similar ground light absorption.
3. The authors state that "the existence of insulating PAEN in active layers does not inhibit charge generation or induce additional charge recombination". They should provide a more detailed experiment explanation.
4. Figure 3g, the detailed test method should be given.
5. "Upon light exposure, the Vout of the amplifier outputs a voltage signal and transmits it to the NFC, thereby recording the electrophysiological signals." I did not see the role of NFC in any Figure or video in the article. In my opinion, the device uses the positive and negative terminals of the SourceMeter to collect signals in the video. In addition, the author should label all the parts in the video.
6. The highlights and innovativeness of the article need to be strengthened.
7. The operating environment of the device is the external environment, yet Figure 3 presents the light stability of the device under glove box conditions when testing its light stability. Discussing the light stability of the encapsulated device in the external environment, compared to the glove box environment, may have more practical application value and reference significance.
8. The authors have applied this device to the recording of electrophysiological signals, which is a highly potential application area. On this basis, have the authors considered conducting in-depth analysis of the recorded signals to extract useful information closely related to physiological processes? For example, when analyzing electromyograms, can they infer the movement of a certain muscle group, thereby potentially deducing the posture of the limb?
9. There are some formatting errors in the text, Page 12, where "N2-filled" should be subscripted in the Figure 3d, f. The same problem occurs on page 24, and the authors are supposed to correct it. In addition, the label "looking down" in Figure 5e has been misspelled as "looking dawn", and the authors should double-check the manuscript to make sure that all the textual representations are accurate.

Version 1:

Reviewer comments:

Reviewer #1

(Remarks to the Author)

I am satisfied that all of my comments have been addressed.

Reviewer #2

(Remarks to the Author)

The reviewer acknowledges that the authors have made notable improvements in addressing several comments from the initial review. However, the manuscript continues to fall short in key areas that are essential for demonstrating a truly autonomous and wearable biosensing system.

A critical issue remains unresolved: while the original manuscript proposed a self-powered wireless biosensor incorporating an NFC communication module, the revised version omits this feature entirely. Instead of demonstrating wireless data transmission, the authors now rely on a commercial digital multimeter for signal acquisition. Although technical limitations are cited in the rebuttal, the removal of the NFC component significantly undermines the novelty and practical impact of the work.

One of the central claims of this study is the development of a skin-attachable, fully self-sustained biosensing platform. Such a system should integrate both power autonomy and wireless data communication. While the authors have made commendable progress on the power supply aspect through optimized organic solar cell (OSC) performance, the lack of integrated wireless communication prevents the realization of a truly standalone wearable device.

Furthermore, the overall form factor of the system remains bulky, as acknowledged by the authors. This compromises its suitability for practical wearable applications and limits its translational relevance.

In conclusion, although the study presents valuable scientific contributions at the materials and device levels, it does not convincingly demonstrate a fully integrated, autonomous, and practical wearable biosensing platform. The absence of wireless communication and the bulky form factor significantly limit the novelty and real-world applicability of the work.

Reviewer #3

(Remarks to the Author)

The authors have addressed my comments and the paper is recommended for publication.

Reviewer #1:

In this manuscript, Ma and co-workers reported a wearable self-powered bio-sensor that features a unipolar amplifier consisting of a dual-OECTs and powered by the OSC. Regarding the power supply unit, the author incorporated BS3TSe-4F as guest acceptor and an insulating poly(aryl ether) (PAEN) into the PM6/BTP-eC9, which system achieved better power and operational stabilities. For the bio-sensor module, the unipolar amplifier based on p(g2T-T) shows low-power and better performance compared with the single OECT device. As a result, the author demonstrated that this self-powered integrated device shows more stable physiological signal output and faster response compared to bio-sensors based on single OECT, across a wide range of illumination intensities (500 lux-50,000 lux).

In this integrated device level, the authors highlighted the simple architecture for wearable low-power self-powered bio-sensors without external energy supplies/storage, which are crucial for ensuring the long-term and reliable function of self-powered biosensors in diverse environmental conditions. I think this manuscript is scientifically interesting and technological important and would be one of the highly important research articles in this journal. However, there are some critical concerns that must be addressed, and the authors need to carefully revise the manuscript.

1. The author selected insulating poly (aryl ether) (PAEN) material as additives to construct the better performance of OSCs. I suspect that the film absorption of PAEN has a certain effect on the device performance, such as J_{sc} , but I have not found the relevant data. I suggest that the authors should also provide the film absorption of PAEN.

Response: We are grateful for the reviewer's thoughtful questions and constructive suggestions.. We apologize for the confusion of the reviewer's logic due to our lack of the film absorption of PAEN, and we have added the absorption spectrum of PAEN in **Fig. R1b**. As shown in **Fig. R1**, the film absorption of PAEF is located at 200-350 nm and is completely transparent in the visible range. This result indicates that there is no obvious change in the absorption spectra after blending PAEN with PM6/BTP-eC9:BS3TSe-4F in the range of 400-1000 nm (**Fig. R1c**). Therefore, the effect of PAEF absorption on the device short-current is negligible.

We added the film absorption of PAEN in the revision manuscript and marked in red.

Fig. R1 (Supplementary Fig. 3) | a Normalized absorption spectra of the pristine films of active layer materials. **b** Normalized absorption spectra of the PAEN and glass substrate. **c** Normalized absorption spectra of the different film devices based on sequential layer-by-layer processing technology.

2. PAEN is a type of an insulating polymer material, so why is device performance not affected? Please explain it.

Response: Thank you very much for this comment. Indeed, PAEN is a type of an insulating polymer material. But this material was used as the third component to construct high performance OSCs (**ACS**

Energy Lett. 2022, 7, 2927; *Adv. Funct. Mater.* 2020, 30, 2003654; *Cell Rep. Phys. Sci.* 2021, 2, 100408.). According to the previous results, the dipole moment of PAEN is 16.49 D, which is much greater than those of PM6 (1.78 D) and Y6 (1.25 D) (*ACS Energy Lett.* 2022, 7, 2927). Generally, a high dipole moment can enhance the dielectric properties of the materials, which could decrease charge recombination and the exciton binding energy by the surrounding dielectric environment. Herein, the dielectric constant (ϵ_r) at 1 MHz for PAEN is 4.5, which is greater than the one ($\epsilon_r \approx 2$) for PM6. Although the addition of PAEN into the PM6/BTP-eC9:BS3TSe-4F system, containing acceptor with remarkable small exciton binding energies, did not show significantly improved charge generation and decay dynamics in the TA spectra (**Fig. 2i**), such high dipoles and surrounding high dielectric environment could also be beneficial for the high photovoltaic performance. In addition, to investigate the charge recombination probabilities of the host binary and the optimized ternary devices, we have studied the exciton dynamics, the charge generation, transport, and the collection processes in the corresponding optimized devices by employing steady-state and transient spectroscopic techniques (**Fig. 2 and Supplementary Figs. 6-9**). Meanwhile, a thorough morphological study can be helpful to understand the effect of PAEN in the corresponding active layer films (**Supplementary Figs. 10-12**). In general, the systematic investigation of the physical mechanism and morphological characterization in the corresponding active layer films implies that PAEN can effectively improve the performance of OSCs, which is also consistent with the previous conclusions (*ACS Energy Lett.* 2022, 7, 2927; *Adv. Funct. Mater.* 2020, 30, 2003654).

3. The authors should provide the standard deviation of PCE, such as Table S3.

Response: Thank you very much for this suggestion. We modified it in the revised manuscript and marked in red.

4. As OECT sensor are also illuminated by light together with OSC, the authors should demonstrate the channel current of OECT sensor will be significantly influenced by light.

Response: We thank the reviewer for this insightful question. As you suggested, we supplemented experiments to verify the channel current of OECT sensor will be significantly influenced by light. As shown in **Fig. R2**, we measured the transfer curves of SMU supplied OECT sensors at different light intensity. The transfer curves show negligible difference, indicating the channel current and threshold voltage will not be influenced by light illumination.

Fig. R2 (supplemented data) | Transfer curves of a SMU supplied OECT sensor measured at different light intensity.

5. In Figure 4h, the authors showed that the amplifier can be very stable during the long-term operation. The stability of the single OECT and related discussion should also be provided here.

Response: Thank you very much for the these comments and suggestions. We supplemented experiments to evaluate the stability of OSC powered single OEECT. As shown in **Fig. R3**, the OEECT maintained 88% of its initial peak on-state current after 4150 s light illumination (50,000 lux) and electrochemical cycling, probably due to the Redox side reactions or material structure evolution. However, the slight performance degradation does not influence the stability of the amplifier (please see **Supplementary Fig. 17**), whose performance is not very sensitive to the peak on-state current of the OEECTs.

We added the stability of OSC powered single OEECT in the revision manuscript and marked in red, as follow: “On the contrary, the single OEECT powered by the same OSCs maintained 88% of its initial peak on-state current after 4150 s light illumination (50,000 lux) and electrochemical cycling, probably due to the Redox side reactions or material structure evolution (**Supplementary Fig. 19**).”

Fig. R3 (Supplementary Fig. 19) | Operation stability of an OSC powered single OEECT. a Circuit diagram used for the test. **b** Operation stability of the OSCs powered single OEECT with an input frequency of 1 Hz. The light intensity used here is 50,000 lux.

- This manuscript is coherently written, but some mistakes need further correction, as follows: 1) In abstract section, this sentence “We show that this integrated self-power bio-sensor....., across a wide range of (500 lux-50,000 lux) illumination intensities.” is corrected as “We show that this integrated self-power bio-sensor....., across a wide range of illumination intensities (500 lux-50,000 lux).” 2) Photostability or light stability of these words, which is described stability. Please keep the manuscript consistent. 3) “single OEECT” is not “single OEECTs”. Please carefully check and correct all errors in this manuscript.

Response: Thank the reviewer for pointing out this mistake. We have modified it in the revised manuscript and marked in red.

Reviewer #2:

The manuscript presents wearable organic electrochemical transistor (OECT)-based biosensors powered by organic semiconductors (OSCs). The study highlights a dual-OECT configuration that achieves stable signal output and faster response times for electrophysiological signal recording. Additionally, the authors report flexible OSCs with a power conversion efficiency (PCE) of 19.04% and robust operational stability. Despite these achievements, the work falls short in demonstrating a fully integrated wearable electronic system. The lack of comprehensive system-level demonstrations limits the practical applicability and impact of the proposed technology. Furthermore, the manuscript does not present significant advancements in materials, fabrication processes, device configurations, or performance metrics for either OSC or OECT technologies. These limitations prevent the work from meeting the innovation and impact criteria expected for publication in Nature Communications. Regrettably, the reviewer cannot recommend this manuscript for publication in Nature Communications. Detailed comments and constructive feedback are provided below for the authors' consideration.

1. The presented work lacks an innovative approach to device integration, a critical aspect for wearable technologies. The fabrication of OSCs on a plastic substrate and OECTs on a glass substrate using disparate techniques such as evaporation and spin-coating contradicts the concept of a seamlessly integrated wearable device. A single-substrate approach is essential for the high-level integration expected in modern wearable electronics, which the manuscript fails to deliver. The lack of an actual system-level demonstration, including the integration of OSCs, OECTs, and NFC components, makes it difficult to assess the feasibility of the entire system as a wearable device. Images or schematics showing the complete integrated system are necessary to evaluate the practical implementation of the described technologies.

Response: We appreciate your valuable suggestions regarding our manuscript and your interest in previously published research in the field. In response to your comments, we would like to elucidate how our study contributes to existing knowledge and advances the field.

- 1) According to the previous reports, the major challenges for wearable bio-sensors using organic electrochemical transistors (OECTs) powered by flexible organic solar cells (OSCs) include flexibility, energy autonomous and stable signal output without external energy storage devices. OSCs suffer from significant power and voltage losses as light intensity decreases, along with poor stability (light and mechanical stabilities), their unstable autonomous energy without external energy supplies leads to unstable signal output of high-power bio-sensors based on single OECT. Specifically, in applications involving single OECTs, small variations in the gate bias can cause significant changes in the output drain current, thereby affecting the stability of electrophysiological signal outputs. Under unstable autonomous energy of OSCs, these fluctuations become more pronounced, potentially leading to device failure. While complex power managers systems are adept at smoothing power output of solar cells under fluctuating conditions and maintaining the operational stability of wearable bioelectronics, their integration needs to be strategically dealt with to avoid increasing the size and complexity drastically. In a systematic level, innovations in system design must continuously aim at reducing the physical footprint and optimizing the energy flow management to bolster functionality across varying environmental settings. Therefore, morphological optimization is central to addressing the fundamental challenges of OSCs' power and operational stability. Advancements in a system-level design of low-power OECT sensors is equally essential to ensure minimal voltage fluctuation, particularly in designs that eschew bulky energy storage modules. Above innovations are crucial for ensuring the long-term and reliable function of self-powered biosensors in diverse environmental conditions.

In this work, we report a wearable self-powered low-power unipolar amplifier designed to address

the challenges of maintaining consistent performance in wearable biosensors, particularly under varying light conditions, completely eliminating the need for external energy storage modules. This low-power amplifier features a dual-OECTs configuration, connected in series and powered by champion OSCs, enabling the continuous and reliable monitoring of human electrophysiological signals under ambient lighting.

- a) For the power supply unit, we aim to address the problem of low output power and poor stability (light and mechanical stabilities) across a wide range of illumination intensities for wearable use. Introducing BS3TSe-4F into the PM6/BTP-eC9 host blend can optimize energy offsets and molecular packing of the host materials and improve the photovoltaic performance. In addition, insulating PAEN material has highly twisted-stiff backbones without any side chains, which can enhance mechanical stability of active layer. Meanwhile, the dipole moment of PAEN is 16.49 D, which is much greater than those of PM6(1.78 D) and acceptor. Generally, a high dipole moment can enhance the dielectric properties of the materials, which could decrease charge recombination and the exciton binding energy by the surrounding dielectric environment. As a result, the introduction of BS3TSe-4F and PAEN into PM6/BTP-eC9 system (PM6/PAEN/BTP-eC9:BS3TSe-4F) not only exhibits maximum performance, but also shows the better light and mechanical stabilities. Note of that the optimized OSCs is one of the highest performance values of flexible devices (**Supplementary Fig. 13b**).
- b) For the bio-sensor component, we designed the low-power unipolar amplifier, which features a dual-OECTs configuration, connected in series. For this kind of circuit, the trip point does not depend on the supply voltages that supplied by the OSC. System characterization demonstrates that the unipolar amplifier operates with a maximum gain of 93 (V/V) near $V_{in} = 0$ V and it shows low-power compared with the single OECT device. Meanwhile, this unipolar amplifier also solves the problem of complex preparation of bipolar OECT devices.
- c) In a systematic level, we integrated the OSCs and amplifier in a single-substrate manner to reduce the cost of integrated device size and power loss. In order to validate the effective application of the integrated devices on-body use, we have done quantitative analyses of signal-to-noise ratio (SNR) under different application conditions (**Supplementary Fig. 24**). For the different light intensity, the optimized OSC-powered amplifier consistently delivered high SNR and signal quality across multiple electrophysiological measurement scenarios - whether under day, dawn or night conditions - substantially outperforming the single OECT configurations. Furthermore, we extensively evaluated its long-term electrical and mechanical stability, along with other non-ideal factors such as motion artifacts. The optimized OSC-powered amplifier device shows no SNR decay after one week, regardless of activity state, confirming the device's stability and minimal noise generation during monitoring. In summary, our research establishes that the champion OSC-powered amplifier delivers stable and reliable electrophysiological signal recording with minimal light intensity dependence, paving the way for self-powered wearable biosensors suitable for indoor, point-of-care health monitoring, and Internet of Things applications.

We thank the reviewer for this question and recognize that these differences may not have been made sufficiently clearly in the manuscript before. We have improved the manuscript, especially in the “Results” section. We hope that this clarification addresses your concerns about the novelty of our study. Thank you for your valuable time and conditions, and we remain open to any further suggestions or feedback.

- 2) Our integrated devices are processed separately on the same flexible substrate instead of two different substrates, as described in the **Fig. 1d** and **Supplementary Fig. 16**. This single-substrate self-powered sensors were separately fabricated by a spin coating process. In the following work, we will pattern the integrated devices by using the blade-coating or slot-die coating process to

reduce the process complexity caused by the spin coating process. In addition, we apologize for the lack of an actual system-level demonstration and thanks for the reviewer's suggestions. In response to the reviewer's request for system-level validation, we devoted considerable effort to developing a mobile application for portable bioelectrical signal monitoring. However, technical constraints inherent in current integrated circuit technology and software development frameworks precluded full realization of this component. We sincerely regret that we were unable to fulfill this aspect of our original proposal. To substantiate the applicability of our integrated sensing platform, we implemented an alternative validation protocol using a precision digital multimeter as the signal acquisition terminal. Although this transitional methodology did not completely attain the envisioned system integration goals, it nevertheless yielded reliable empirical data that effectively validated the prototype's operational efficacy in biomedical signal monitoring applications.

In our revised manuscript, we added the **Supplementary Video1-500 lux** and **Video2-5000 lux** of a system-level demonstration and marked different components in the **Supplementary Fig. 21**.

Fig. R4 (Supplementary Fig. 21) | The capture of Video S1 that shows the ECG monitoring at 500 lux, right panel shows the different components of integrated devices. The manuscript does not clarify the power management between the OSCs and the NFC chip. It is crucial to specify whether the NFC is self-powered by the OSCs or relies on external power sources, as this impacts the device's autonomy and practicality in wearable applications.

Response: We thank the reviewer for this comment. We apologize for the confusion caused by our failure to specify the power supply of NFC. NFC (Near Field Communication) wireless power supply is realized by integrating the two functions of NFC communication and wireless charging. NFC uses a reference frequency of 13.56 MHz, uses NFC communication links to control transmission, and sends a continuous carrier signal to achieve power transmission to the NFC tag, and then establishes a communication channel to powering the device. For example, in a wireless charging scenario, when an NFC-enabled device (such as a mobile phone) is close to a charging pad or base with a built-in NFC tag chip, the NFC tag communicates with the device and the device provides the required power. NFC communication has the advantage of wireless charging, so we chose NFC communication to reduce the power management system design of integrated devices. Thanks again to the reviewers for their attention.

In response to the reviewer's requirements for system-level verification, we dedicated substantial resources and collaborated with external stakeholders to develop a portable mobile application for bioelectric signal monitoring. However, due to inherent technical limitations in current integrated circuit technology and software frameworks, full realization of this component was not achievable. We sincerely regret this unanticipated outcome and apologize for the removal of the NFC communication module from the original proposal. To objectively demonstrate the core functionality of our integrated

device, we implemented an alternative verification protocol using a benchtop digital multimeter (DMM) as a validation platform. This approach allowed us to confirm critical operational parameters while maintaining rigorous experimental standards. We acknowledge the inconvenience caused by this modification and commit to pursuing long-term solutions for wireless communication implementation in future iterations.

- Concerns about the physical and operational scalability of the organic electrochemical transistors, given their relatively large size, are not addressed. The rationale behind the chosen device dimensions and the impact on overall device performance and application potential should be critically analyzed.

Response: We thank the reviewer for this question about device size. Indeed, we also agree that the OECT devices in current stage are still bulky. This kind of device does not rely on the complicated and high-cost photolithography technique to achieve high pattern resolution. It is well-known that the large device dimension may retard the device response speed. As you suggested, we analyzed the influence of device dimension to the device response speed. **Fig. R5 b-c** show the amplification capability of an OSC powered amplifier ($W=1000\ \mu\text{m}$, $L=50\ \text{nm}$, $d=35\ \text{nm}$) at 167 Hz (higher than the main frequency of ECG: 1-2 Hz), EEG: 3-100 Hz and EMG: 20-150 Hz signals). This integrated device can still provide a gain value higher than 10 V/V. Therefore, we can conclude that the device dimension will not significantly influence the application of our device in the self-powered electrophysiological recording. Even so, future works can still focus on scaling down the integrated device to achieve higher operation frequency and integration density.

Fig. R5 | Amplification capability of OSC powered amplifier at 167 Hz (light intensity: 50,000 lux). **a** Circuit diagram used for the test. **b** Transient output response of an OSC powered amplifier ($W=1000\ \mu\text{m}$, $L=50\ \text{nm}$, $d=35\ \text{nm}$) upon small triangular input. **c** the transient gain of the OSC powered amplifier.

- The experimental results, particularly those related to on-body applications, lack comprehensive quantitative analysis. Data on the signal-to-noise ratio and detailed frequency analysis are necessary to substantiate the claims of effective physiological signal monitoring. There is no sufficient evidence provided on the biocompatibility and long-term stability of the device when in contact with human skin. These are crucial factors for any wearable technology, especially those intended for continuous health monitoring.

Response: We are grateful for the reviewer's thoughtful questions and constructive suggestions. As you suggested, we supplemented the following experiments and data analysis:

- For quantitative analysis, we calculated the signal-to-noise ratio ($\text{SNR}=20\log(\text{peak}/\text{STD})$, where 'Peak' and 'STD' denote the peak amplitude and standard deviation of the noise voltage/current, respectively) of the electrophysiological signals recorded by OSC powered amplifier and single OECT. As shown in **Figs. R6a-b** and **Table R1**, the electrophysiological signals recorded by OSC powered

amplifier shows higher SNR and less light intensity dependency when compared with its single OECT counterpart. Besides, we also compared the SNR of our devices with that of other reported

O
E
C
T

We added the signal-to-noise ratio (SNR) analyses and the relative calculation method in the revision manuscript and marked them in red, as follow: “To quantitatively evaluate the signal quality of integrated devices under varying illumination conditions, we conducted signal-to-noise ratio (SNR) analyses (Methods section). As illustrated in Figs. 5c-e and Supplementary Table 10, the OSC-powered amplifier demonstrates superior performance in electrophysiological recordings, exhibiting both higher SNR values and reduced light-intensity dependency compared to single OECT devices. For

i
r
n
s
s
t
.
a
n
s

$$SNB(\text{dB})=20 \log(\text{Peak}/\sigma)$$

Where ‘Peak’ denotes the average peak voltage/current amplitude of the signals. ‘ σ ’ denotes the standard deviation of the noise voltage/current at 0 V input.”.

h
i
n
E
M
G
r
e
c
o
r
d
i
n
g
s
,
t
h
e
a
m
p
l
i
f
i
e
r
m

Fig. R6 | Signal-to-noise ratio (SNR) of the electrophysical signals recorded by OSC powered amplifier and single OECT. a SNR of EMG. **b** SNR of ECG. **c** SNR of EOG. **d** Summary of the SNR of ECG signals recorded by other OECT-based sensors.

Table R1 (Supplementary Table 10) | Signal-to-noise ratio (SNR) of the electrophysical signals recorded by OSC powered amplifier and single OECT.

	Light intensity		
	500 lux	5,000 lux	50,000 lux
SNR of EMG (OSC powered single OECT)	21.1±4.3 dB	27.3±0.9 dB	35.0±2.8 dB
SNR of EMG (OSC powered amplifier)	41.7±1.6 dB	44.3±2.2 dB	44.5±2.9 dB
SNR of ECG (OSC powered single OECT)	29.6±2.4 dB	31.0±2.2 dB	34.1±4.9 dB
SNR of ECG (OSC powered amplifier)	43.0±4.0 dB	43.0±1.5 dB	43.9±3.7 dB
SNR of EOG (OSC powered single OECT)	20.9±4.5 dB	28.3±2.1 dB	42.7±1.9 dB
SNR of EOG (OSC powered amplifier)	50.0±0.7 dB	51.2±1.0 dB	54.0±4.7 dB

Table R2 Summary of the SNR of ECG signals recorded by other OECT-based sensors.

Device and materials	SNR of ECG (dB)	Power (μ W)	Ref.
OSC powered single OECT (PEDOT:PSS)	25.9	0	Nature 2018, 561, 516.
OECT (PEDOT:PSS)	24	~300	Adv. Funct. Mater. 2019, 29, 1906982.
OECT (PEDOT:PSS)	21.7	~150	Adv. Mater. Technol. 2023, 8, 2200611.
Solid-state OECT (PEDOT:PSS)	32.5	~450	Adv. Funct. Mater. 2023, 33, 2209354.
OECT (PEDOT:PSS)	29	~180	Adv. Funct. Mater. 2022, 32, 2200458
OECT (PEDOT:PSS)	52	--	Sci. Adv. 2018, 4: eaau2426
OECT (PEDOT:PSS)	12.2	~120	ACS Appl. Mater. Interfaces 2022, 14, 24840.
Ion-gated organic electrochemical transistor (PEDOT:PSS)	53.86	~600	Nat. Mater. 2020, 19, 679.
OECT (PEDOT:PSS)	20	~400	Chem. Eng. J. 2024, 483, 148980.
OSC powered OECT amplifier (p(g2T-TT))	43.0±4.0	0	Our work

- 2) For biocompatibility, we supplemented cell culture experiment on both OECT and OSC parts. In this experiment, NCTC clone 929 (L-929) cells were cultured on the exposed p(g2T-T)-based OECT channels, PDMS encapsulated OSCs and bare glasses (control) for 1, 3 and 5 days. **Fig. R7a**

shows the live (green)-dead (red) fluorescent staining photographs. We can find that both OECT and OSC parts show much higher cell density than control. Besides, as shown in **Fig R7c**, cells on OECT and OSC parts shows higher mitochondrial activity than control. Therefore, we can demonstrate the good biocompatibility and safe cell-device interfaces of our all-organic integrated system.

We added the biocompatibility analyses and the relative experiment method in the revision manuscript and marked them in red, as follow: “Before the on-body recording, we firstly assessed the biocompatibility of the integrated device using the L-929 cell-culture experiment (Method Section). As shown in **Supplementary Fig. 20**, both OECT channel and PDMS encapsulated OSC part show higher cell density and mitochondrial activity than control samples (bare glass). This indicates that the good biocompatibility and safe cell-device interfaces can be achieved using our all-organic integrated system. Taking advantages of the robust performance and high biocompatibility of this self-powered bio-sensors, we then carried out electrophysiological signal recording under different surrounding environment.”.

Fig. R7 (Supplementary Fig. 20) | Biocompatibility test of different parts in the integrated device. **a** Fluorescence microscope of the PC-12 cells cultured on different device-cell interfaces. Cell density **b** and 3-(4,5-Dimethylthiazol-2-yl)-2,5-diphenyltetrazolium bromide (MTT) tested mitochondrial activities **c** on different device-cell interfaces.

3) For long-term stability, we supplemented on-body stability test in air atmosphere. As shown in **Fig. R8**, an OSC powered amplifier device was attached on the body of a health volunteer. Output curves and gain were recorded after different days. We found that the gain value only slightly dropped from 42 V/V to 32 V/V on the first day. After that, the device showed good stability. Besides, this on-body integrated device was also used for long-term ECG recording. As shown **Fig. R8d**, clear ECG signals can be captured from 1 to 7 days. These results demonstrate the good long-term stability of our integrate device, which origins from the stability of both OECT amplifier part (**Fig. 4h**) and OSC part (**Fig. 3**).

We added the long-term on-body stability data and related analysis in the revision manuscript and marked them in red, as follow: “To validate practical implementation, we extensively evaluated

its long-term electrical and mechanical stability, along with other non-ideal factors such as motion artifacts. **Supplementary Fig. 22** illustrates the device's long-term on-body stability over one week (168h), revealing only a slight decrease in gain from 42 V/V to 32 V/V on the first day, after which it stabilizes. Additionally, the device was employed for long-term ECG monitoring, successfully capturing high-SNR ECG signals from day 1 to day 7 (**Supplementary Fig. 22d**).

Fig. R8 (Supplementary Fig. 21) | Long-term stability of an on-body OSC powered amplifier in air atmosphere (light intensity: 5,000 lux). Output curve **a** and gain curve **b** of the OSC powered amplifier after different storage time. Inset shows the on-body OSC powered amplifier used for test. **c** Plot of gain values versus storage time. **d** ECG signals recorded by OSC powered amplifier on different days.

4. Authors need to adequately address the stability of the device under real-world conditions, including its mechanical and electrical stability under repeated stress and strain, which are typical in wearable scenarios.

Response: We thank the reviewer for this suggestion. We tested the electrical performance of an OSC powered amplifier before and after 500 repeated bending cycles. As shown in **Fig. R9**, no obvious performance deterioration was found during the test. The good mechanical stability originates from the good mechanical stability of both p(g2T-T) (*Science* 2024, 386, 431-439) and OSC materials (**Figs. 3g-i**).

We added the mechanical stability data and related analysis in the revision manuscript and marked them in red, as follow: “Mechanical stability, a critical factor for flexible devices, was also tested. As depicted in **Supplementary Fig. 22**, the device exhibited no significant performance degradation after 500 bending cycles (radius: 5 mm). These findings underscore the excellent long-term stability of our integrated device, attributable to the robust OECT amplifier and OSC (**Figs. 3c-i**) modules.”

Fig. R9 (Supplementary Fig. 23) | Mechanical stability of the OSC powered amplifier.

- Noise considerations, particularly those arising from motion artifacts, are not discussed. The stability of device adhesion on a PET substrate during actual use and its effect on signal integrity need thorough examination and robust testing data.

Response: We thank the reviewer for this insightful question about noise analysis. To analyze the influence from 1) motion artifacts and 2) stability of device adhesion on substrate, we tested the SNR of the ECG signals recorded by an OSC powered amplifier in rest and jogging conditions during the long-term on-body wearing.

For motion artifacts. As shown **Fig. R10b-c**, in jogging state, motion artifacts indeed reduce the SNR from 39.9 ± 0.5 dB to 33.0 ± 10.5 dB. However, the ECG signal is still clear and its SNR is still higher than that of the signal recorded by OSC powered single OECT (30.7 ± 2.2 dB). We believe this noise can be eliminated in the future by using adhesive electrolyte such as hydrogel.

For stability of device adhesion on substrate. As shown **Fig. R10b-c**, we found that the SNR does not decay after one week of on-body wearing, no matter in rest or jogging state. In all cases, SNR is higher than that of the signals recorded by OSC powered single OECT (30.7 ± 2.2 dB). This indicates that the device adhesion stability is good, which will not produce noise during the monitoring.

Therefore, we added the related analysis in the revision manuscript and marked them in red, as follow: “Additional noise characterization studies addressed potential degradation mechanisms and motion artefacts, with the system showing particular resistance to such interference factors. The SNR of ECG signals recorded by the OSC-powered amplifier was assessed during both rest and jogging states (**Supplementary Fig. 24a**). Regarding device degradation, **Supplementary Figs. 24b-c** shows no SNR decay after one week (168h), regardless of activity state, confirming the device's stability and

m
i
n
i
m
a
l

n
o
i
s
e

g
e
n
e

Fig. R10 (Supplementary Fig. 24) | Noise analysis of signals recorded by integrated devices (light intensity: 5,000 lux). a ECG signals recorded in rest or jogging state before and after one week of wearing. **b** Signal-to-noise ratio (SNR) of ECG signals recorded at different state before and after one week of wearing.

Reviewer #3:

In this work, they have developed a low-power self-powered physiological sensor that features a dual-OECTs configuration, connected in series and powered by the optimized champion OSCs. The devices employ the efficient and more stable OSCs to power amplifier by suppressing charge recombination and improving flexibility, facilitating long-term, on-demand use. In addition, the integrated self-powered biosensor can be attached to human skin for stable monitoring of physiological signals, including ECG, EMG, and EEG, over a wide range of (500 lux-50,000 lux) illumination intensities. There are some important issues need to clarify/discuss before it can be considered for publication in Nature Communication. These are:

1. In this work, the advantages of PM6, (SM-NFAs)-BTP-eC9 and BS3TSe-4F being selected are not shown.

Response: Thank you very much for this comment. For the OSCs, the PM6:BTP:eC9 system has excellent performance and has been widely applied in the morphology optimization and mechanism analysis of organic solar cells. Therefore, we selected the PM6:BTP:eC9 system as the host to construct better PCE and stability (light and mechanical stabilities). Firstly, BTP-eC9 and BS3TSe-4F have very similar molecular structures, the more selenium-substituted core and π -bridge achieve a slightly enhanced electron-donating ability to give a redshift for BS3TSe-4F. As shown in **Supplementary Fig. 3**, the introduction of BS3TSe-4F into BTP-eC9 can broaden acceptor materials absorption range. In addition, the introduction of BS3TSe-4F as guest into PM6/BTP-eC9 system provides the complementary and broadened absorption compared to that of the PM6:BTP-eC9 system, which can improve photovoltaic performance, especially the J_{SC} . Secondly, the energy cascades for the PM6, BTP-eC9 and BS3TSe-4F are conducive to charge extraction and transport (**Table R2** and **Supplementary Fig. 2**). And then, we used water and ethylene glycol (EG) to conduct surface energy measurements to demonstrate the miscibility of active layer. The calculated Flory-Huggins interaction parameter (χ) values are 0.385 for PM6/BTP-eC9, 0.028 for PM6/BTP-eC9:BS3TSe-4F, and 0.00025 for BTP-eC9:BS3TSe-4F, respectively. The results imply that BTP-eC9 and BS3TSe-4F prefer to form an alloy-like phase in the ternary blend, making the pure phase more crystalline without damaging the scale of phase separation (*Nat. Rev. Mater.* 2019, 4, 229; *Joule* 2021, 5, 2408.). Lastly, we also explored the light stability of the corresponding devices tested, which are a key indicator for measuring the application of a wearable self-powered device. As shown in **Figs. 3c-f**, the PM6/PAEN/ BTP-eC9:BS3TSe-4F device exhibits inferior light stability to that of the PM6/BTP-eC9 device. On the whole, the combination, termed LbL-Ternary/PAEN (PM6/PAEN/BTP-eC9:BS3TSe-4F), not only demonstrates superior photovoltaic performance with a PCE reaching 19.17% - surpassing the 18.02% PCE of PM6/BTP-eC9 (LbL-Binary) device-but also enhances stability (light and mechanical stabilities).

Table R2 | Optical properties of BS3TSe-4F and BTP-eC9

Materials	λ_{max} film (nm)	LUMO (eV)	HOMO (eV)
BTP-eC9	830	-3.94	-3.96
BS3TSe-4F	848	-5.59	-5.66
PM6	614	-3.56	-5.47

2. The authors believe that “introducing BS3TSe-4F as guest into PM6:BTPeC9 system provides the complementary and broadened absorption under the ambient light compared with that of the PM6:BTP-eC9 system.” However, this opinion is inconsistent with the results in Figure S3. Apparently, BS3TSe-4F and BTPeC9 have similar ground light absorption.

Response: Thank you for this comment. Normalized absorption spectra of the pristine materials films

and blends films are shown in **Supplementary Fig. 3**. BS3TSe-4F and BTP-eC9 display maximum absorption peaks at 830 nm and 848 nm, respectively. The more selenium-substituted achieves a slightly enhanced electron-donating ability to give an ≈ 18 nm redshift for BS3TSe-4F, which is consistent with the previous report (*Adv. Mater.* 2022, 34, 2202089). Compared with BTP-eC9, the redshift of 18 nm in the absorption range of BS3TSe-4F is quite large. In addition, the blended system with the addition of BS3TSe-4F also exhibits a redder absorption range compared to PM6/BTP-eC9 (LbL-Binary), demonstrating that our strategy is feasible (**Supplementary Fig. 3b**). Therefore, the BS3TSe-4F as a third component can broaden the absorption range to improve device performance (**Fig. 2** and **Table 1** in the revised manuscript).

Table R2 | Optical properties of BS3TSe-4F and BTP-eC9

Materials	λ_{\max} film (nm)
BTP-eC9	830
BS3TSe-4F	848

- The authors state that “the existence of insulating PAEN in active layers does not inhibit charge generation or induce additional charge recombination”. They should provide a more detailed experiment explanation.

Response: We are grateful for the reviewer's suggestion.. Indeed, PAEN is a type of an insulating polymer material. But this material was used as the third component to construct high performance OSCs (*Cell Rep. Phys. Sci.* 2021, 2, 100408; *ACS Energy Lett.* 2022, 7, 2927; *Adv. Funct. Mater.* 2020, 30, 2003654.). According to the previous results, the dipole moment of PAEN is 16.49 D, which is much greater than those of PM6 (1.78 D) and Y6 (1.25 D) (*ACS Energy Lett.* 2022, 7, 2927). Generally, a high dipole moment can enhance the dielectric properties of the materials, which could decrease charge recombination and the exciton binding energy by the surrounding dielectric environment. Herein, the dielectric constant (ϵ_r) at 1 MHz for PAEN is 4.5, which is greater than the one ($\epsilon_r \approx 2$) for the polymer donor PM6. Although addition of PAEN into the PM6/BTP-eC9:BS3TSe-4F system, containing acceptor with remarkable small exciton binding energies, did not show significantly improved charge generation and decay dynamics in the TA spectra (**Fig. 2i**), such high dipoles and surrounding high dielectric environment could also be beneficial for the high photovoltaic performance. In addition, to investigate the charge recombination probabilities of the host binary and the optimized ternary devices, we have studied the exciton dynamics, the charge generation, transport, and the collection processes in the corresponding optimized devices by employing steady-state and transient spectroscopic techniques (**Fig. 2** and **Supplementary Figs. 6-9**). Meanwhile, a thorough morphological study can be helpful to understand the effect of PAEN in the corresponding active layer films (**Supplementary Figs. 10-12**). In general, the systematic investigation of the physical mechanism and morphological characterization in the corresponding active layer films implies that PAEN can effectively improve the performance of OSCs, which is also consistent with the previous conclusions (*ACS Energy Lett.* 2022, 7, 2927; *Adv. Funct. Mater.* 2020, 30, 2003654).

- Figure 3g, the detailed test method should be given.

Response: Thank you very much for this suggestion. For the strain-stress curve test, we added the detailed test conditions into the Methods section (**Stability Measurements**), as follows: “Pseudo free-standing tensile test: For the tensile testing specimen, the active layers were spin-coated onto the PEDOT:PSS/glass substrate. The active layer specimen with a size of 1.2×0.5 cm was prepared by using a cutting plotter. To float the specimen on the water surface, water was allowed to penetrate into the PEDOT:PSS layer. Subsequently, PEDOT:PSS was dissolved, and the active layer was delaminated from the glass substrate. By performing this process at the water surface, the floating active layer specimen could be obtained. Specimen gripping was achieved by attaching PDMS-coated Al grips on

the specimen gripping areas using van der Waals adhesion. The tensile test was performed by a linear stage with a strain rate of 0.01 mm/s. All of tensile tests were carried out under the ambient conditions (Temperature ~ 25 °C, relative humidity (RH) ~ 30 %).”

5. “Upon light exposure, the Vout of the amplifier outputs a voltage signal and transmits it to the NFC, thereby recording the electrophysiological signals.” I did not see the role of NFC in any Figure or video in the article. In my opinion, the device uses the positive and negative terminals of the SourceMeter to collect signals in the video. In addition, the author should label all the parts in the video.

Response: Thank you for these comments and suggestions. We apologize for the lack of an actual system-level demonstration and thanks for the reviewer’s suggestions. In order to better showcase the integrated system, we dedicated substantial resources and collaborated with external stakeholders to develop a portable mobile application for bioelectric signal monitoring. However, due to inherent technical limitations in current integrated circuit technology and software frameworks, full realization of this component was not achievable. We sincerely regret this unanticipated outcome and apologize for the removal of the NFC communication module from the original proposal. To objectively demonstrate the core functionality of our integrated device, we implemented an alternative verification protocol using a benchtop digital multimeter (DMM) as a validation platform (**Supplementary Video1-500 lux and Video2-5000 lux**). This approach allowed us to confirm critical operational parameters while maintaining rigorous experimental standards. We acknowledge the inconvenience caused by this modification and commit to pursuing long-term solutions for wireless communication implementation in future iterations.

Fig. R9 (Supplementary Fig. 21) | The Video1-500lux capture shows the different components of integrated devices.

6. The highlights and innovativeness of the article need to be strengthened.

Response: We appreciate your valuable suggestions. In response to your comments, we would like to elucidate how our study contributes to existing knowledge and advances the field.

According to the previous reports, the major challenges for wearable bio-sensors using organic electrochemical transistors (OECTs) powered by flexible organic solar cells (OSCs) include flexibility, energy autonomous and stable signal output without external energy storage devices. OSCs suffer from significant power and voltage losses as light intensity decreases, along with poor stability (light and mechanical stabilities), their unstable autonomous energy without external energy supplies leads to unstable signal output of high-power bio-sensors based on single OECT. Specifically, in applications involving single OECTs, small variations in the gate bias can cause significant changes in the output

drain current, thereby affecting the stability of electrophysiological signal outputs. Under unstable autonomous energy of OSCs, these fluctuations become more pronounced, potentially leading to device failure. While complex power managers systems are adept at smoothing power output of solar cells under fluctuating conditions and maintaining the operational stability of wearable bioelectronics, their integration needs to be strategically dealt with to avoid increasing the size and complexity drastically. In a systematic level, innovations in system design must continuously aim at reducing the physical footprint and optimizing the energy flow management to bolster functionality across varying environmental settings. Therefore, morphological optimization is central to addressing the fundamental challenges of OSCs' power and operational stability. Advancements in a system-level design of low-power OEET sensors is equally essential to ensure minimal voltage fluctuation, particularly in designs that eschew bulky energy storage modules. Above innovations are crucial for ensuring the long-term and reliable function of self-powered biosensors in diverse environmental conditions.

In this work, we report a wearable self-powered low-power unipolar amplifier designed to address the challenges of maintaining consistent performance in wearable biosensors, particularly under varying light conditions, completely eliminating the need for external energy storage modules. This low-power amplifier features a dual-OEETs configuration, connected in series and powered by champion OSCs, enabling the continuous and reliable monitoring of human electrophysiological signals under ambient lighting.

- 1) For the power supply unit, we aim to address the problem of low output power and poor stability (light and mechanical stabilities) across a wide range of illumination intensities for wearable use. Introducing BS3TSe-4F into the PM6/BTP-eC9 host blend can optimize energy offsets and molecular packing of the host materials and improve the photovoltaic performance. In addition, insulating PAEN material has highly twisted-stiff backbones without any side chains, which can enhance mechanical stability of active layer. Meanwhile, the dipole moment of PAEN is 16.49 D, which is much greater than those of PM6(1.78 D) and acceptor. Generally, a high dipole moment can enhance the dielectric properties of the materials, which could decrease charge recombination and the exciton binding energy by the surrounding dielectric environment. As a result, the introduction of BS3TSe-4F and PAEN into PM6/BTP-eC9 system (PM6/PAEN/BTP-eC9:BS3TSe-4F) not only exhibits maximum performance, but also shows the better light and mechanical stabilities. Note of that the optimized OSCs is one of the highest performance values of flexible devices (**Supplementary Fig. 13b**).
- 2) For the bio-sensor component, we designed the low-power unipolar amplifier, which features a dual-OEETs configuration, connected in series. For this kind of circuit, the trip point does not depend on the supply voltages that supplied by the OSC. System characterization demonstrates that the unipolar amplifier operates with a maximum gain of 93 (V/V) near $V_{in} = 0$ V and it shows low-power compared with the single OEET device. Meanwhile, this unipolar amplifier also solves the problem of complex preparation of bipolar OEET devices.
- 3) In a systematic level, we integrated the OSCs and amplifier in a single-substrate manner to reduce the cost of integrated device size and power loss. In order to validate the effective application of the integrated devices on-body use, we have done quantitative analyses of signal-to-noise ratio (SNR) under different application conditions (**Supplementary Figs. 19-22**). For the different light intensity, the optimized OSC-powered amplifier consistently delivered high SNR and signal quality across multiple electrophysiological measurement scenarios - whether under day, dawn or night conditions - substantially outperforming the single OEET configurations. Furthermore, we extensively evaluated its long-term electrical and mechanical stability, along with other non-ideal factors such as motion artifacts. The optimized OSC-powered amplifier device shows no SNR decay after one week, regardless of activity state, confirming the device's stability and minimal

noise generation during monitoring. In summary, our research establishes that the champion OSC-powered amplifier delivers stable and reliable electrophysiological signal recording with minimal light intensity dependence, paving the way for self-powered wearable biosensors suitable for indoor, point-of-care health monitoring, and Internet of Things applications.

We thank the reviewer for this comment and recognize that these differences may not have been made sufficiently clearly in the manuscript before. We have improved the manuscript, especially in the “Results” section. We hope that this clarification addresses your concerns about the novelty of our study. Thank you for your valuable time and conditions, and we remain open to any further suggestions or feedback.

7. The operating environment of the device is the external environment, yet Figure 3 presents the light stability of the device under glove box conditions when testing its light stability. Discussing the light stability of the encapsulated device in the external environment, compared to the glove box environment, may have more practical application value and reference significance.

Response: Thank you for these comments and suggestions. We have the following response:

- 1) In fact, the light stability of the integrated device in external air environment was already characterized. **Fig. R11 (Fig. 4h)** shows the operation stability of the OSC powered amplifier in air atmosphere with strong light intensity (50,000 lux LED). The device maintains 100% of its initial V_{out} after 4150 s of illumination and electrochemical redox cycle. This data demonstrates that both OSC and amplifier part are very stable in strong light and air environment.
- 2) To further investigate the long-term stability of the integrated device in air and ambient light environment, we supplemented on-body stability test. As shown in **Fig. R12**, an OSC powered amplifier was attached on the body of a health volunteer. Output curves and gain were recorded after different days. We found that the gain value only slightly dropped from 42 V/V to 32 V/V on the first day. After that, the device showed good stability. Besides, this on-body device was also used for long-term ECG recording. As shown **Fig. R12d**, clear ECG signals can be captured from 1 to 7 days. These results demonstrate the good long-term stability of our integrate device, which origins from the stability of both OECT amplifier part (**Fig. R11**) and OSC part (**Figs. 3c-f**).

We added the long-term on-body stability data and related analysis in the revised manuscript and marked them in red, as follow: “Additional noise characterization studies addressed potential degradation mechanisms and motion artefacts, with the system showing particular resistance to such interference factors. The SNR of ECG signals recorded by the OSC-powered amplifier was assessed during both rest and jogging states (**Supplementary Fig. 24a**). Regarding device degradation, **Supplementary Figs. 24b-c** shows no SNR decay after one week (168h), regardless of activity state, confirming the device's stability and minimal noise generation during monitoring. For motion artifacts,

w
h
i
l
e

t
h
e

S
N
R

d
e
c

Fig. R11 (Fig. 4h) | Operation stability of the OSCs powered amplifier with an input frequency of 1 Hz. The light intensity used here is 50,000 lux

Fig. R12 (Supplementary Fig. 22) Long-term stability of an on-body OSC powered amplifier in air atmosphere (light intensity: 5,000 lux). a On-body OSC powered amplifier used for test. **b** ECG signals recorded in rest or jogging state before and after one week of wearing. **c** Signal-to-noise ratio (SNR) of ECG signals recorded at different state before and after one week of wearing.

8. The authors have applied this device to the recording of electrophysiological signals, which is a highly potential application area. On this basis, have the authors considered conducting in-depth analysis of the recorded signals to extract useful information closely related to physiological processes? For example, when analyzing electromyograms, can they infer the movement of a certain muscle group, thereby potentially deducing the posture of the limb?

Response: Thank you for these comments and suggestions. Indeed, an in-depth analysis of the signals recorded by integrated devices can help to determine the physical behavior or to prevent physiological diseases. This is a very important project for specific applications of wearable devices. In the future, we believe that this integrated system could be integrated with human-machine interface (HMI) or

Virtual Reality (VR) technology for facilitate easier observation and judgement, which may be beyond the scope of the present work. We thank the reviewers again for their attention to this work and our will continue to investigate it in the future.

9. There are some formatting errors in the text, Page 12, where “N₂-filled” should be subscripted in the Figure 3d, f. The same problem occurs on page 24, and the authors are supposed to correct it. In addition, the label “looking down” in Figure 5e has been misspelled as “looking dawn”, and the authors should double-check the manuscript to make sure that all the textual representations are accurate.

Response: We thank the reviewer for pointing out this mistake. We modified it in the revised manuscript and marked in red.

Reviewer #1 (Remarks to the Author):

I am satisfied that all of my comments have been addressed.

Response: We are truly grateful to the reviewers for their attentive examination and helpful suggestions, which have significantly improved our work.

Reviewer #2 (Remarks to the Author):

The reviewer acknowledges that the authors have made notable improvements in addressing several comments from the initial review. However, the manuscript continues to fall short in key areas that are essential for demonstrating a truly autonomous and wearable biosensing system.

A critical issue remains unresolved: while the original manuscript proposed a self-powered wireless biosensor incorporating an NFC communication module, the revised version omits this feature entirely. Instead of demonstrating wireless data transmission, the authors now rely on a commercial digital multimeter for signal acquisition. Although technical limitations are cited in the rebuttal, the removal of the NFC component significantly undermines the novelty and practical impact of the work.

One of the central claims of this study is the development of a skin-attachable, fully self-sustained biosensing platform. Such a system should integrate both power autonomy and wireless data communication. While the authors have made commendable progress on the power supply aspect through optimized organic solar cell (OSC) performance, the lack of integrated wireless communication prevents the realization of a truly standalone wearable device.

Furthermore, the overall form factor of the system remains bulky, as acknowledged by the authors. This compromises its suitability for practical wearable applications and limits its translational relevance.

In conclusion, although the study presents valuable scientific contributions at the materials and device levels, it does not convincingly demonstrate a fully integrated, autonomous, and practical wearable biosensing platform. The absence of wireless communication and the bulky form factor significantly limit the novelty and real-world applicability of the work.

Response: We sincerely appreciate the reviewers' recognition of our efforts in material and device integration. Their comments are highly relevant and insightful. We acknowledge that the current system would benefit from a more comprehensive system-level demonstration to better highlight the integration and autonomous operation of the wearable biosensing platform. We are also grateful for the reviewer' s valuable suggestion regarding the use of NFC technology. However, considering the current limitations of NFC in terms of signal stability and practical applicability, we have adopted a Bluetooth-based communication solution to ensure reliable performance and overall system robustness. It is important to note that the choice of communication protocol is not the core innovation of this work, which instead focuses on the self-powered sensor design. We hope for the reviewer' s understanding regarding this design choice. In response to the suggestion, we have now included a system demonstration (**Video S1** and **Video S2**) with detailed annotations to more clearly illustrate the operation of the integrated platform.

Furthermore, the size of the current self-powered sensor module (1.5 cm × 2 cm) suggest potential for further optimization in miniaturization. It should be noted that the present stage is a laboratory-scale demonstration, and this integrated device serves as a foundational platform for subsequent commercialization. Future work will focus on miniaturizing the self-powered system and pursuing enhanced NFC solutions to more fully realize the ideal of self-powering.

As you suggested, we added the use of self-powered wearable integrated devices in the revision manuscript and marked them in red, as follow: “Upon light exposure, the V_{out} of the amplifier outputs a voltage signal, which is transmitted via Bluetooth to a mobile phone for electrophysiological signal recording (**Supplementary Video** and **Supplementary Fig. 21**)”

Fig. R1 (Supplementary Fig. 21) | The capture of Video S1 that shows the ECG monitoring at 500 lux right panel shows the different components of integrated devices.

In addition, we also added the use of self-powered wearable integrated devices in the revision manuscript (Methods Section) and marked them in red, as follow: “The ECG measurement setup was built around a circuit board, which housed the key components: an ADS1292 chip for signal acquisition, singlechip (STM321051) and a E104-bt5005A Bluetooth low-energy module for communication (3.8V Li-ion electronic power supply). The connection between this board and the self-powered biosensor is established through flexible electrodes and connecting wires (**Supplementary Fig. 21**). ECG signals are streamed in real time to a mobile device (Redmi 14C) for display within a dedicated app.”

Reviewer #3 (Remarks to the Author):

The authors have addressed my comments and the paper is recommended for publication.

Response: We are truly grateful to the reviewers for their attentive examination and helpful suggestions, which have significantly improved our work.

Reviewer #2 (Remarks to the Author):

Response: We are truly grateful to the reviewers for their attentive examination and helpful suggestions, which have significantly improved our work.